# Locomotory behaviour of early tetrapods from Blue Beach, Nova Scotia, revealed by novel microanatomical analysis

Kendra I. Lennie[1,2], Sarah L. Manske[2,4], Chris F. Mansky[5] and Jason S. Anderson[2,3]

[1]Biological Sciences, University of Calgary, 507 Campus Drive NW, Calgary, Alberta, Canada T2N 1N4
[2]McCaig Institute for Bone and Joint Health, and [3]Comparative Biology and Experimental Medicine, Foothills Campus, University of Calgary, 3330 Hospital Drive NW, Calgary, Alberta, Canada T2N 4N1
[4]Radiology, Foothills Medical Centre, University of Calgary, 1403-29th Street NW, Calgary, Alberta, Canada T2N 2T9
[5]Blue Beach Fossil Museum, 127 Blue Beach Road, Hantsport, Nova Scotia, Canada B0P 1P0

KIL, 0000-0002-6614-925X

**Subject Areas:**
palaeontology/evolution/biomechanics

**Keywords:**
tetrapods, Carboniferous, locomotion, Devonian, fin-to-limb, water-to-land

**Author for correspondence:**
Kendra I. Lennie
e-mail: kendra.lennie@ucalgary.ca

Evidence for terrestriality in early tetrapods is fundamentally contradictory. Fossil trackways attributed to early terrestrial tetrapods long predate the first body fossils from the Late Devonian. However, the Devonian body fossils demonstrate an obligatorily aquatic lifestyle. Complicating our understanding of the transition from water to land is a pronounced gap in the fossil record between the aquatic Devonian taxa and presumably terrestrial tetrapods from the later Early Carboniferous. Recent work suggests that an obligatorily aquatic habit persists much higher in the tetrapod tree than previously recognized. Here, we present independent microanatomical data of locomotor capability from the earliest Carboniferous of Blue Beach, Nova Scotia. The site preserves limb bones from taxa representative of Late Devonian to mid-Carboniferous faunas as well as a rich trackway record. Given that bone remodels in response to functional stresses including gravity and ground reaction forces, we analysed both the midshaft compactness profiles and trabecular anisotropy, the latter using a new whole bone approach. Our findings suggest that early tetrapods retained an aquatic lifestyle despite varied limb morphologies, prior to their emergence onto land. These results suggest that trackways attributed to early tetrapods be closely scrutinized for additional information regarding their creation conditions, and demand an expansion of sampling to better identify the first terrestrial tetrapods.

# 1. Significance statement

It is thought that terrestriality emerged between the Late Devonian, represented by the early tetrapod *Ichthyostega* [1,2] and the middle Early Carboniferous, exemplified by *Pederpes* [3,4]. Recently, multiple studies have reconsidered locomotor behaviour in these and other early tetrapods by examining patterns of external morphology, and from this reconstructing soft tissue and possible biomechanical ability [1,3,5,6]. These studies make conclusions about terrestrial capabilities based on potential mobility developed from external bone features rather than direct evidence derived from locomotion during life, a limitation recently noted by Dickson *et al.* [6]. We first use a previously established cross-sectional compactness profile analysis of the femora of the Blue Beach tetrapods to infer aquatic or terrestrial lifestyles [7,8]. Next, we describe a novel method using first-order evidence from microanatomy of the trabecular bone within the medullary cavity across a whole femur, rather than second-order inferences based off a theoretical reconstruction of soft tissue. Such analysis across an entire bone has never been done before; previous trabecular bone analyses select areas of trabecular bone expected to convey a signal and, therefore, may miss large-scale or unexpected patterns in trabecular bone. To resist habitual stress and strain (loading) on a limb bone, the internal trabecular structure becomes highly organized through remodelling in a non-homogeneous manner [9–14]. Our method shows that vertebrates moving across terrestrial environments experience patterns of stress reflected in internal limb bone features due to gravity and ground reaction forces. From this, we can infer whether an animal did walk on land, not that it had the potentiality. By contrast, we observed that vertebrates moving in a semi-aquatic environment do not show patterns in trabecular structure consistent with these gravitational forces. Examination of internal structures of limb bones provides more robust data to test hypotheses of the locomotory capacity of early tetrapods because these structures are less affected by external wear and erosion and represent an independent, direct source of data for locomotor history of an organism [9,13,14].

# 2. Introduction

The evolution of digited limbs once appeared intimately tied with the transition from water to land, but now the water-to-land and fin-to-limb transitions are no longer assumed to be synonymous [15–17]. Early inferences about lifestyles of early tetrapods were often drawn from cranial features (i.e. lack of lateral line canal grooves) in conjunction with the presence of 'well-developed' or 'robust' limbs [18,19]. Recently, soft tissue reconstruction and biomechanics have become common methods used to infer which early tetrapod was the earliest candidate capable of walking on land [1,3,6,20–22]. The use of the extant phylogenetic bracket (EPB) [23,24], which is essential (even if in a modified form) [3,21,22] to propose a hypothetical reconstruction of soft tissue, depends on a poorly resolved early tetrapod phylogeny [3]. Efforts to reconstruct soft tissues are hampered by, as Bryant & Seymour [25] observed, the fact that a great deal of muscle matter does not leave reliable osteological correlates on bone in living animals. Using osteological correlates (in conjunction with the EPB), therefore, may not lead to a complete inference of soft tissue location or orientation in fossil material and thus removes some constraint from the biomechanical model thereby derived [3,20,21]. Furthermore, biomechanical inferences based on hypothesized soft tissue reconstructions only suggest that a given animal might have been capable of some range of motion, but provide little information on probable range of motion [1,6] or actual use. In 2012, the limb joint mobility of *Ichthyostega* was modelled [1] and the authors concluded the animal had potential terrestrial capabilities, based on the possible range of motion. A more recent study [6] used tetrapod humeral shape and traits, and possible load-bearing capabilities based on external morphology to predict when tetrapods were first capable of terrestrial locomotion. Here, we ask, do the fossils show evidence of responding to gravity and ground reaction forces, thus directly demonstrating terrestrial behaviour, rather than inferring its possibility?

Comparison with modern taxa, especially modern aquatic taxa, presents challenges even to more robust analyses into locomotor mode of early tetrapods. The first is that most modern aquatic vertebrates are secondarily aquatic, so controlling for primary or secondary aquatic nature is difficult. Additionally, many modern aquatic animals have bone microstructure adaptations that are closely associated with counteracting buoyancy associated with well-developed mammalian lungs [26,27]. Other bone microstructure features may be associated more closely with phylogeny, ontogeny or allometry [8,11,26]. For this reason, it is best to examine a variety of features, of which we are increasingly able to extract and visualize in further detail with the accessibility of modern microcomputed tomographical (µCT) techniques.

Compactness profiles from histological sections of long bones are capable of giving direct insight into behavioural stresses influencing bone formation and absorption [7,28–30]. However, a single two-dimensional image may not capture where and how forces may be passing through limb bones. Studies on mammals and birds have shown that long bone trabecular structure is sensitive to differing loading patterns, and trabeculae align with mechanical loading of the limb bone [13,14]. Acquiring such data from early tetrapod limb bones would be invaluable in reconstructing locomotory behaviours and patterns in the earliest vertebrates to transition from an aquatic to terrestrial lifestyle.

In our new whole bone approach, we explore qualitatively and quantitatively three-dimensional orientations of trabecular bone from which we can infer if, where, and in which plane, the internal structure of a limb bone was remodelling in response to habitual movement under the influence of gravity and ground reaction forces.

Palaeoanthropology has a rich literature on the quantification of three-dimensional bone structure in extant and extinct mammalian material [9,31–41], which shows that behaviourally moderated repetitive forces acting on bone induce bone remodelling [9,11,12,42,43]. Therefore, the organization of trabecular remodelling provides additional and potentially more sensitive information about forces having acted on the bone in relation to the animals' locomotory behaviour than hypothetical muscle reconstructions. Kivell [9] effectively summarizes that (i) trabecular bone remodels throughout life in a more predictable way than external bone features; (ii) trabecular bone (in conjunction with cortical bone) reflects actual behaviours (locomotor strategies in the context of the present study), rather than potential behaviours; and (iii) there does not appear to be an allometric relationship between the size of the animal and degree of anisotropy. However, mechanical loading effects on internal structures of load-bearing bones have only recently been approached in non-mammalian fossils. As of 2019, only two studies have examined such data; one on dinosaurs [12] and another on non-avian reptiles [11].

One of the earliest post-Devonian sites within the Early Carboniferous worldwide hiatus in the fossil record is Blue Beach in Nova Scotia, Canada. It preserves a diverse fauna with elements from both the Late Devonian and later Carboniferous tetrapod faunas as well as a rich trackway record [44–46]. These conditions make it an excellent locality to test the terrestrial capabilities of taxa along the fin-to-limb transition. Here, we take advantage of bone's response to stresses and the three-dimensional preservation of the isolated limb elements to analyse internal bone structure for a signature of terrestrial locomotion. Using a combination of two-dimensional midshaft analyses and three-dimensional patterns of trabecular alignment from µCT, we compare midshaft variables of the fossils with those of animals of known lifestyles to test patterns of trabecular anisotropy. Finally, we provide quantitative variables and discuss them in comparison with animals of known locomotory strategy.

## 2.1. Blue Beach

Blue Beach, located near Hantsport, Nova Scotia, preserves the type section of the Horton Bluff Formation, dated to the early-mid to mid-Tournaisian [47], and is located on the shore of an estuary off the Bay of Fundy, where the heavy tidal action frequently exposes new fossil material [46]. At the time of deposition, Blue Beach was a marginal, marine-influenced environment, ranging from onshore, aerially exposed horizons that supported lycopod forests, to offshore facies described as lagoonal [40,43] and often stormy [46]. These conditions explain the presence of aquatic, semi-aquatic and possibly terrestrial vertebrates at the same locality. Trackways suggest the presence of terrestrial taxa [45,46].

Blue Beach preserves several elements that are similar to Devonian taxa *Acanthostega* and *Tulerpeton* [44], which have been previously described as aquatic, based on the presence of internal gills in *Acanthostega* [17] and the aquatic depositional setting of the *Tulerpeton* material [48]. Blue Beach also has a diverse array of animals similar to material known from the Early Carboniferous [44,46], such as a community of whatcheeriid-like animals including both aquatic (*Whatcheeria*-like) [49] and more terrestrial (*Pederpes*-like and *Ossinodus*-like; for discussion on whatcheerid grade animals, see [50]) tetrapods [4,44,50,51]. Other Carboniferous morphotypes include elements comparable to more amphibious *Eoherpeton* [52] and *Proterogyrinus* [19]. The elements are found in isolation but are excellently preserved in three dimensions, enabling informative internal analyses.

We conducted a series of analyses on limb elements belonging to as great a taxonomic range as possible. The greatest number of a single element is the femur, which we used for greatest consistency in comparison; taxonomic identification and previously interpreted lifestyles are shown in table 1.

**Table 1.** Blue Beach fossil femora taxonomic identifications and hypothesized locomotory behaviours.

| specimen | element | comparable to | previous locomotor behaviour association |
|---|---|---|---|
| NSM.005.GF.045.043 | femur | embolomere | semi-aquatic to terrestrial |
| RM.20.6711 | femur | acanthostegid | aquatic |
| NSM.007.GF.004.630 | femur | tulerpetontid | aquatic |
| NSM.005.GF.045.044 | distal femur | whatcheeriid (*Ossinodus*-like) | semi-aquatic to terrestrial |
| NSM.005.GF.045.047 | femur | whatcheeriid (*Ossinodus*-like) | semi-aquatic to terrestrial |
| YPM.PU.20103 | femur | whatcheeriid (revised) | aquatic to semi-aquatic |

µCT scans revealed intact internal structures that contrasted well with the surrounding matrix, which enabled us to obtain clear midshaft slices and analyse trabeculae in three dimensions.

## 2.2. Compactness profiles

Both cortical and trabecular bone respond to physiological and mechanical stresses: in cortical bone, responses include thickening or thinning of the bone [10,53], whereas in trabecular bone, there is remodelling of trabecular alignment to better withstand mechanical forces transmitted through the bone [9,31,54]. Recent palaeohistological investigations reveal patterns of internal anatomy in the limb bones of extant and extinct tetrapods, including thickness of cortical bone and long bone compactness, can often (but not always) be associated with various patterns of limb use [8,11,26,28,29,55,56]. In extant tetrapods, a compact cortical bone, with a distinct medullary cavity and clear transition between cortical and trabecular bone are often associated with limb bones capable of effective terrestrial locomotion, whereas a less compact cortical layer, gradual transition from cortical to trabecular bone, and trabeculae occupying the medullary cavity are often associated with an aquatic lifestyle [8,11,26,29,55]. These microanatomical features observed in cross-section vary depending on the location of the sampled section within the bone and which bone is sampled [26]. Variation within a single bone is controlled for by using a mid-diapheseal section [26,55]. There are little data on the microanatomy of femoral midshafts of extant aquatic animals, in part due to the reduction of hindlimbs common in secondarily aquatic limbed vertebrates [26]. This study, therefore, relies on midshafts of semi-aquatic tetrapods for a limited comparative sample.

We studied the compactness profiles of eight extant tetrapods (three terrestrial, three aquatic and two semi-aquatic) and five femoral fossils from Blue Beach using a previously published method [7,8]. Bone Profiler [7] reads a two-dimensional image of the midshaft of a long bone and quantifies patterns in relative cortical and trabecular location and thicknesses. Here, we used a µCT image from the mid-diaphysis (midshaft) as described by Canoville & Laurin [55].

## 2.3. Bone functional adaptation

Wolff's Law, a phenomenon of trabecular organization first observed by Roux [57] and erroneously attributed to Wolff, describes how trabecular bone remodels based on the influence of habitual stress and strain on the bone [58,59]. Anisotropy is the non-random (organized) alignment of a three-dimensional structure. Patterns in anisotropy of trabecular bone have been linked to patterns of use/behaviours of limbs and their extremities in medical [31,36,39,40] and palaeoanthropological studies [37,43,60]. Analyses of the three-dimensional anisotropy of trabecular bone, in conjunction with bone volume fraction (BV/TV), are often used to make inferences about bone functional adaptation (BFA), and thus, behaviours involving repetitive force in extinct animals [32,37,38,42,43]. Trabecular structure was observed to remodel in alignment with the direction of repetitive mechanical loading forces in sheep [13,14]. For this study, we use the principles of BFA—bones remodel to better respond to repetitive loading forces—to look for signs of a response to gravitational and ground reaction forces, and thus terrestriality, in the Blue Beach tetrapod femora.

## 2.4. Anisotropy and bone volume fraction

Behaviourally moderated repetitive forces acting on bone induce bone remodelling [9,11]. The organization of trabeculae, therefore, may provide more sensitive information about forces acting on the bone than a two-dimensional midshaft analysis. It has been suggested that results pertaining to

trabecular anisotropy (the non-random alignment of a three-dimensional structure) are sensitive to the location of the volume of interest (VOI) within the bone [43,60]. For this reason, we conducted a whole bone analysis of trabecular anisotropy to maximize the likelihood of capturing terrestrial signal across the entire element and find potential local orientations that we would not have sampled choosing a single area *a priori*.

## 2.5. Principal components analysis

To analyse potential similarities and differences more rigorously between known aquatic and terrestrial taxa and the fossils from Blue Beach, we conducted a principal component analysis (PCA). Our analysis used of a combination of Dragonfly (ORS, Montreal) and Bone Profiler variables.

# 3. Material and methods

## 3.1. Abbreviations

Blue Beach Fossil Museum, Hantsport, Nova Scotia, Canada (BBFM); Florida Museum of Natural History (FMNH); Musée d'Histoire Naturelle de Miguasha, Quebec, Canada (MHNM); Museum of Vertebrate Zoology, University of California, Berkeley, CA, USA (MVZ); Nova Scotia Museum of Natural History, Halifax, Canada (NSM); Redpath Museum, McGill University, Montreal, Canada (RM); University of Florida (UF); University of Michigan Museum of Zoology, Ann Arbor, MI, USA (UMMZ); Yale Peabody Museum of Natural History, New Haven, CT, USA (YPM); Museum für Naturkunde Berlin, Germany (ZMB).

Mode of frequency of anisotropy (Ani.M); bone volume (BV); bone volume fraction (BV/TV); total volume (TV); average cortical area (Ct.Ar); average marrow area (Ma.Ar); average total area (Tt.Ar); cortical area fraction (Ct.Ar/Tt.Ar); average cortical thickness (Ct.Th); average trabecular thickness (Tb.Th); proximal to distal length measurement (P–D); maximum asymptote of angular analysis (Max ang or $R_{max}$); minimum asymptote of angular analysis (Min ang or $R_{min}$); transition point in compactness profile of angular analysis (*P* ang); inverse of the slope of the compactness curve at transition point *P* ang (*S* ang); stiffness score (*R/t*).

## 3.2. Scanning

Fossil tetrapod limb bones from Blue Beach were borrowed from YPM, RM, NSM and BBFM. Unprepared or partially prepared material was prepared by K.I.L. at the University of Calgary. *Eusthenopteron* fin material was borrowed from MHNM. All fossil material was scanned using a Xradia Versa 520 (Carl Zeiss, Germany) with settings of 140 kV, 72 µA, 0.4× magnification and no filter. The scan produced stacks of 1928–2412 images approximately 1000 pixels wide by 1024 pixels tall, and voxel sizes ranging between 21.9 and 32.2 µm.

Extant limb bones studied here include the leopard gecko *Eublepharis*, spiny tailed lizard *Uromastyx*, marine iguana *Amblyrhynchus cristatus* (FMNH UF41558), hellbender salamander *Cryptobranchus alleganiensis* (FMNH UF10881), platypus *Ornithorhynchus anatinus* (MVZ MVZ32885), femur of a (juvenile) housecat *Felis* and a femur of the turtle *Elseya dentata* (UMMZ herps 203554). We also sampled a humerus of the extinct tetrapodomorph fish *Eusthenopteron foordi* (MHNM 06-2674A), to ensure that a primarily aquatic limb bone was considered. Unfortunately, we could not obtain a sufficiently preserved *Eusthenopteron* femur. These elements were used to provide a range of material with varying locomotor strategies for comparison with Blue Beach fossils. *Eublepharis*, *Uromastyx* and *Felis* samples were obtained from specimens used for previous research or teaching at the University of Calgary. Limb bones from the University of Calgary were scanned in a Skyscan 1173 (Bruker, Belgium) at the University of Calgary with settings between 80–120 kV and 70–85 µA with a 9.6–30 µm voxel size. Scans of limb bones from https://www.morphosource.org included *Elseya dentata* (UMMZ 203554), *Caretta caretta* (YPM.VZ.002957), *Ornithorhynchus ornithorhynchus anatinus* ssp. (MVZ 32885), *Amblyrhynchus cristatus* (UF41558) and *Cryptobranchus alleganiensis* (UF10881).

Scans of *Elseya dentata* (UMMZ 203554) and *Caretta caretta* (YPM.VZ.002957) were not used for trabecular modelling as the resolution was insufficient to capture subtleties in trabecular structure. Large measurements such as variables in the compactness profiles and Tt.Ar, Ma.Ar, Ct.Ar, TV and BV were obtained (Tr.Th, Ct.Th were not). The internal structure of *Cryptobranchus alleganiensis*

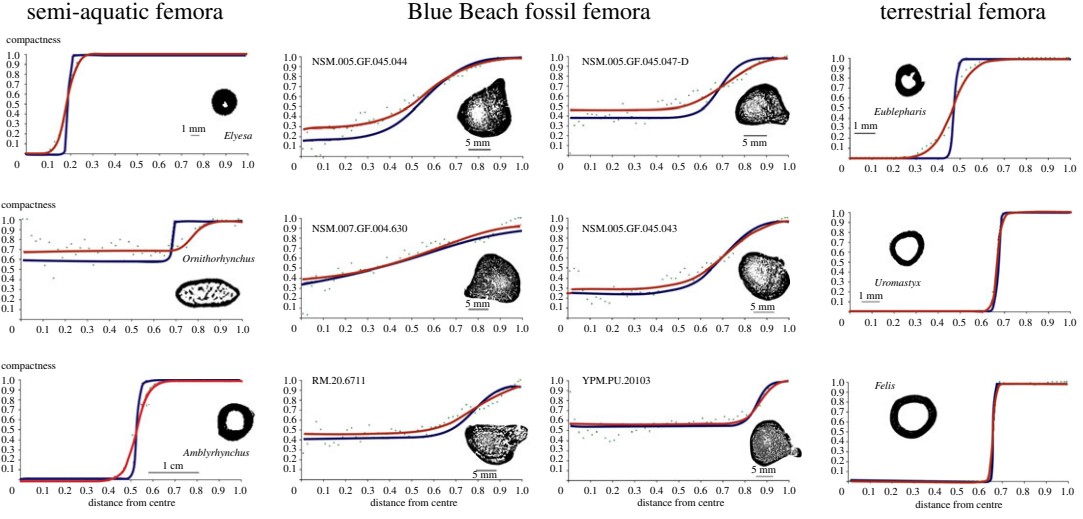

**Figure 1.** Compactness graphs with the corresponding binarized midshaft. All of the fossil femora have non-zero values of compactness at the centre, more similar to the platypus than the cat. The transition from the minimum to maximum asymptote is gradual compared with the abrupt transitions in the platypus and cat. Blue lines reflect angular values; red lines are global. Fossil sample scale bars represent 5 mm; all others represent 1 mm except *Amblyrhynchus* which is 10 mm.

(UF10881) was of a different density than other bone material in the µCT scan, and the internal material was very thick, which meant volumes of interest of the internal material at times only captured the surfaces of the structure, rather than orientations of the structures. An arbitrary but large sample spacing and radius of influence (1500 µm) was used to image a vector map, but no quantitative data were collected from *Cryptobranchus* and it could not be relied upon as a comparative sample.

## 3.3. Two-dimensional analysis

Midshafts were identified by loading the scanned image stack into Data Viewer (Bruker) or Dragonfly (ORS, Montreal), and finding the narrowest circumference when viewed in the sagittal and coronal planes. To ensure the sagittal and coronal planes were aligned with the long axis of the bone, the image stack was additionally loaded in Dragonfly and transverse plane aligned to be at right angles with the sagittal and coronal planes. Once identified, the images in this region were further examined in the transaxial plane to determine which image had the thickest cortical bone. Once selected, the single midshaft image was thresholded using ImageJ (Bethesda, MD, USA). One specimen, YPM.PU.20103, was segmented using Dragonfly (ORS) because ImageJ was unable to autothreshold the image and beam hardening was more easily corrected using Dragonfly. Midshaft images were converted to binary images (figure 1) before being analysed in Bone Profiler. Bone Profiler automatically analysed the exported images and determined the section centre, medullary centre and ontogenetic centre from which global and angular measurements were taken to describe the compactness profile of the bones. Finally, the angular variables of the minimum asymptotes ($R_{min}$) from the compactness profiles of the Blue Beach femora were analysed using a predictive binary formula modelled from a range of aquatic to terrestrial amniote femora. For a detailed description of femoral midshaft analyses and binary formula, see Quemeneur *et al*. [8].

## 3.4. Three-dimensional analysis

We expect a whole bone analysis to detect large-scale patterns in aligned trabecular structure, with terrestrial animals displaying perceptible patterns of non-homogeneous trabecular alignment consistent with transmission of gravitational and ground reaction forces. Additionally, a whole bone analysis may pick up signals from sites of habitually used muscle attachment.

The entire scan of each fossil and extant femur was imported into Dragonfly ORS and the Bone Analysis Application was used to segment cortical and trabecular structures of the whole bone. The Kohler method of automatic segmentation [61] was used because there was more manual control over the differentiation of cortical and trabecular areas, which helped for the unusually shaped specimens.

For the fossil femora, a median filter of kernel size 3 (electronic supplementary material, figure S1) followed by a mean shift filter of 5 were applied (electronic supplementary material, figure S2). The background/air of the scan was thresholded and overwritten so analyses would only be performed on bone and internal bone features and cavities (electronic supplementary material, figure S3). Using the image processing toolbox, an arithmetic filter was used to convert the dataset to a 32-bit float file (electronic supplementary material, figure S4), which was necessary for the later step of applying a distance of Gaussian (DoG) filter. A polynomial of the third degree (electronic supplementary material, figure S5) was applied prior to a DoG with first sigma between 0.5 and 1.0, and second sigma between 2.0 and 4.0 (not normalized; electronic supplementary material, figure S6). The polynomial-DoG dataset was filtered to create a region of interest (ROI) which captured the trabecular structure (electronic supplementary material, figure S8), and the median–mean shift dataset was filtered to create an ROI which more effectively captured cortical bone (electronic supplementary material, figure S9). The two ROIs were merged to create an accurate representation of the cortical and trabecular bone structures (electronic supplementary material, figure S10). The data were then processed as described below.

Cortical and trabecular bone were segmented automatically using the Kohler method (electronic supplementary material, figures S11 and S12). A mask of the whole bone was created by filling the inner area of the cortical bone, which was necessary for computing the measurements. This was achieved by a union of the cortical and trabecular area (not bone) region of interest (electronic supplementary material, figures S13 and S14). The trabecular bone ROI underwent a multi-ROI analysis to remove noise (electronic supplementary material, figures S15 and S16). A box was created around the whole limb bone sampled (electronic supplementary material, figure S17); this was necessary for the program to know where to measure bone volume (BV, TV) and areas (Tt.Ar, Ct.Ar, Ma.Ar) of bone regions and features (Ct.Th, Tr.Th) within the bone. Using the denoised trabecular bone, ROI (electronic supplementary material, figures S16 and S18) sample spacing and radius of influence were approximately 500 µm, or three times the average trabecular width if trabecular width was greater than 167 µm. This resulted in 50% overlap in order to detect patterns in anisotropic areas across a region (electronic supplementary material, figures S18 and S19).

## 3.5. Anisotropy vector data

Dragonfly's bone analysis renders both a vector and scalar map. The vector map provides a rendering of each sample area's direction and value of anisotropy. Here, we used the surface normals algorithm (electronic supplementary material, figures S20 and S21).

## 3.6. Anisotropy scalar data

Qualitative export of data on the frequency of anisotropy was not available at the time of this study; however, histograms of the frequency of anisotropic values throughout each element were created as follows.

The scalar map is used to assess and analyse the mode frequency of anisotropic values in each limb element. First, an ROI of the trabecular area was inverted (to remove noise from segmentation edges; electronic supplementary material, figure S22). The scalar map was converted to nearest neighbour interpolation (electronic supplementary material, figures S23 and S24), and the inverted trabecular area ROI was overwritten with a value of 0 (electronic supplementary material, figures S25–S27), this is why on the frequency histogram values of 0 must be discarded.

## 3.7. Statistics

The calculated values from Bone Profiler and Dragonfly underwent a PCA to provide qualitative results. The PCA was run using a correlation matrix in PAST (for summary and loadings see electronic supplementary material, S2). The impact of phylogeny was not taken into account for this study, and a phylogenetic PCA (pPCA) was not used because of high-level taxonomic identification of Blue Beach specimens, poorly resolved early tetrapod phylogenies and the fact that no living tetrapods are phylogenetically proximate to early tetrapods (see above).

# 4. Results

## 4.1. Compactness profiles

The binary images of the fossil femoral midshafts (figure 1) clearly show the medullary space filled entirely with trabecular bone, suggesting a more aquatic lifestyle. Compactness graphs (figure 1) show that the midshafts of the Blue Beach femora are unlike the graphs of both terrestrial (*Eublepharis, Uromastyx* and *Felis*; figure 1) and semi-aquatic (*Elyesa, Ornithorhynchus* and *Amblyrhynchus*; figure 1) taxa; however, they are more dissimilar to the terrestrial taxa than the semi-aquatic taxa. The semi-aquatic taxa have trabecular bone within the medullary cavity and throughout the transition zone between medullary cavity and cortical bone. The Blue Beach femora have trabecular bone within the medullary cavity, and the transition from medullary cavity to cortical bone is gradual. Additionally, the Blue Beach femora have non-zero compactness values at the bone centre, a trait shared with *Ornithorhynchus*, indicating the presence of trabecular bone through the entire medullary cavity. The Blue Beach femora, which were predicted to represent a range of aquatic to terrestrial taxa based on known trackway and body fossils present at the site (table 1; [44–46]), were all designated aquatic by the binary formula of Quemeneur *et al.* [8].

## 4.2. Trabecular anisotropy

### 4.2.1. Patterns of trabecular structure in modern terrestrial and semi-aquatic tetrapods

Using our whole bone approach (see extended methods; electronic supplementary material, S1) to examine multiple small regions of anisotropy, we show that terrestrial animals do have clear patterns of highly anisotropic trabeculae at the proximal and distal ends of their femora (figure 2). Additionally, *Eublepharis* and *Uromastyx* have highly anisotropic patterning of trabeculae near the midshaft of the bone.

Semi-aquatic vertebrate femora show no pattern in the highly anisotropic regions of trabecular bone at the proximal and distal ends of the femora (figure 3). A case may be made for the femur of *Ornithorhynchus*, which has a greater number of highly anisotropic regions than either *Cryptobranchus* or *Amblyrhynchus*; however, the pattern is not as clearly oriented proximodistally as in *Uromastyx* or *Eublepharis* (figure 2). This could reflect variation in sprawling compared with upright posture, but these regions of high anisotropy match well with topographical muscle maps of the femur of *Ornithorhynchus* (figure 4). Another finding of the whole bone analysis was that qualitatively the frequency of anisotropic regions throughout terrestrial and semi-aquatic extant femora may differ (at the time of this paper's preparation the ability to export the raw data from the histogram was unavailable; however, the developers at Dragonfly are working on it). The peak frequency of anisotropic value in *Uromastyx* and *Felis* is greater than 0.4, and in *Eublepharis*, qualitatively, there is no discernable most common value of anisotropy.

A brief examination of extant femora indicates that gravitational and ground reaction forces promote different patterns of trabecular remodelling between terrestrial and semi-aquatic tetrapods.

### 4.2.2. Patterns of trabecular structure in Blue Beach tetrapod fossil femora

With differing patterns of anisotropy in the trabecular structure of extant terrestrial and semi-aquatic femora observed, the same trabecular analysis was run on a sample of fossil femora representative of the diverse tetrapod fauna at Blue Beach.

Our sample of two complete (figure 5) and three partial femora (figure 6) displayed no discernible pattern to the non-homogeneous alignment of their trabecular structures, suggesting gravitational and ground reaction forces were not passing through the limbs as would be expected of a terrestrial tetrapod. Furthermore, the fossil femora have high frequencies of low values of anisotropy, which is most consistent with the primarily aquatic humerus of *Eusthenopteron*; however, caution is advised when comparing a humeral element with femoral ones. When compared with the extant femora, the Blue Beach material is more consistent with the semi-aquatic tetrapods than terrestrial.

Qualitative observations about patterns and values of anisotropy across the extant and fossil material are summarized in table 2.

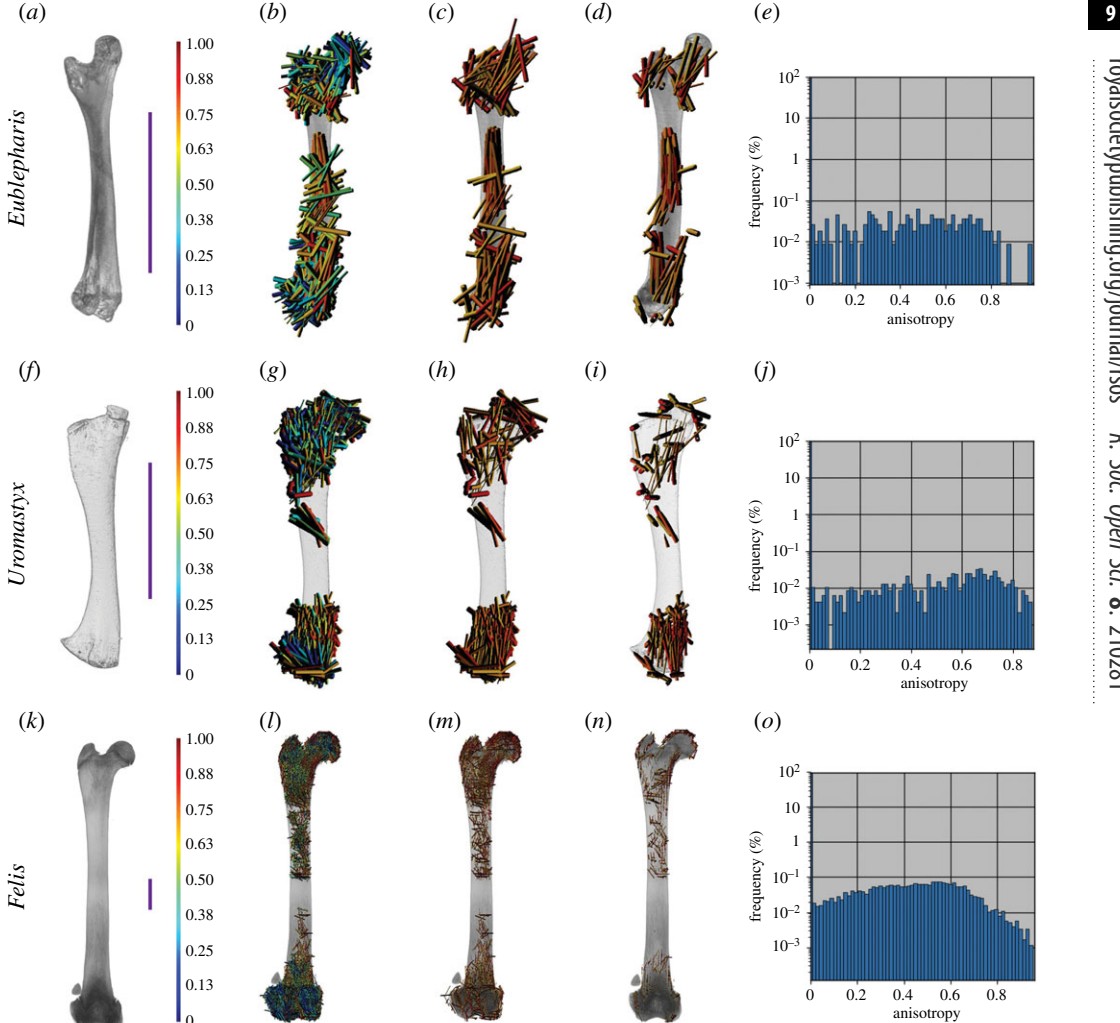

**Figure 2.** Anisotropy vector maps and histograms of extant terrestrial femora. (*a–d*) *Eublepharis* femur. (*b*) Vector map of all anisotropy, (*c*) vector map of anisotropy 0.65 and above, (*d*) vector map of the middle region in the coronal plane with anisotropy 0.65 and above. (*e*) Anisotropy frequency histogram of *Eublepharis* femur. (*f–i*) *Uromastyx* femur. (*g*) Vector map of all anisotropy, (*h*) vector map of anisotropy 0.65 and above, (*i*) vector map of the middle region in the coronal plane with anisotropy 0.65 and above. (*j*) Anisotropy frequency histogram of *Uromastyx* femur. (*k–n*) *Felis* femur. (*l*) Vector map of all anisotropy, (*m*) vector map of anisotropy of 0.65 and above. (*n*) vector map of the mid-region of the bone in the coronal plane with anisotropy 0.65 or higher. (*o*) Anisotropy frequency histogram of *Felis* femur. Scale bars represent 10 mm.

## 4.3. Principal component analysis

PC1 captures variation of the elements due to size and area. PC2 and PC3 capture the variation in bone deposition (i.e. Ct.Th, Tr.Th, Ma.Ar) and variation in observed bone compactness as a function of location of internal bone features in cross-section, and PC4 is heavily influenced by $R_{\min}$. PC2 and PC3 when plotted against PC4 discriminate between terrestrial, semi-aquatic and aquatic locomotion. The Blue Beach femora overlap with other aquatic femora, but not those of semi-aquatic or terrestrial taxa (figure 7).

## 5. Discussion

The fossil femora included animals similar to the aquatic *Tulerpeton* and those believed to be more terrestrial such as *Ossinodus*. All fossil femora are most similar to the extant comparative semi-aquatic animals, but exhibit deviations from the semi-aquatic representative *Ornithorhynchus*. An important caveat to our discussion and conclusions is that the Blue Beach tetrapods represent animals believed to be primarily aquatic (see citations for discussion on the potential presence of secondarily aquatic

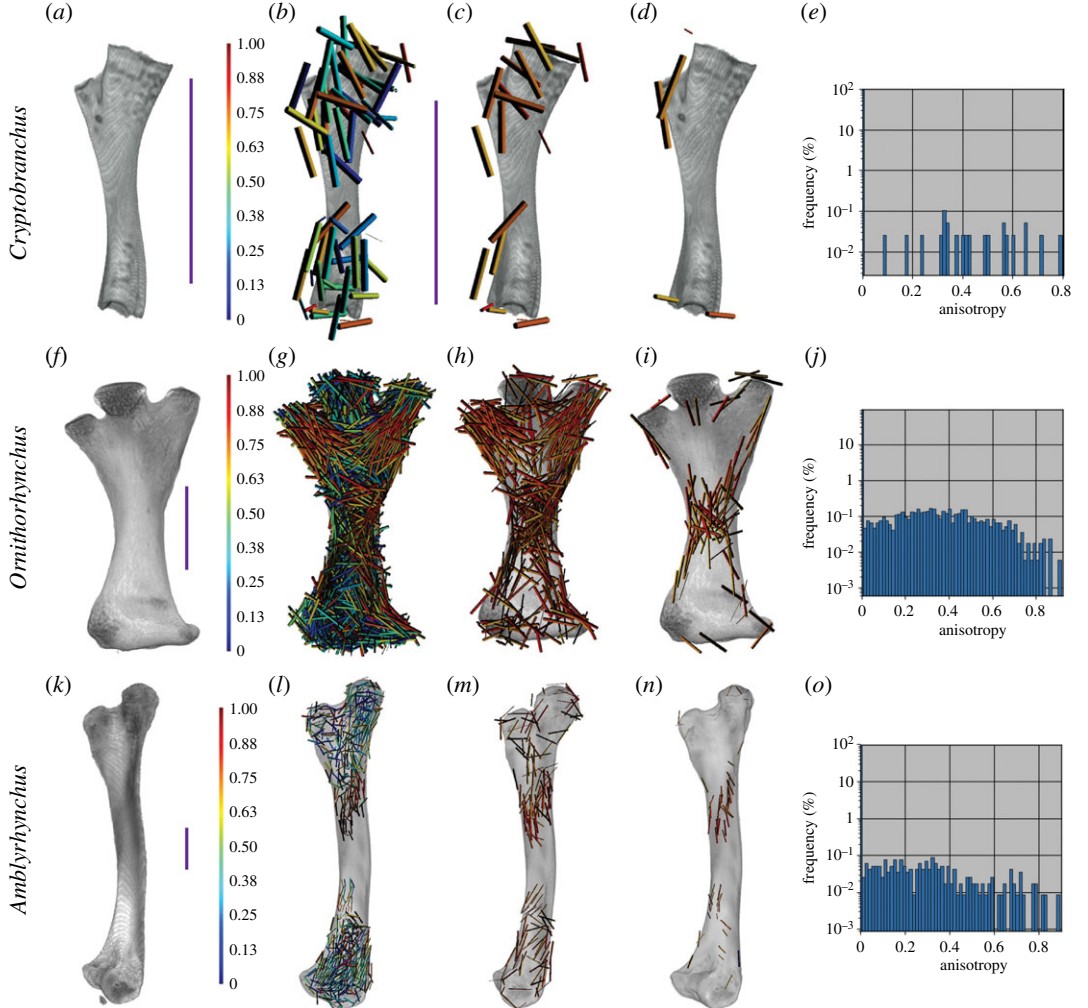

**Figure 3.** Anisotropy vector maps and histograms of extant aquatic and semi-aquatic femora. (*a–d*) *Cryptobranchus* femur. (*b*) Vector map of all anisotropy, (*c*) vector map of anisotropy 0.65 and above, (*d*) vector map of the middle region in the coronal plane with anisotropy 0.65 and above. (*e*) Anisotropy frequency histogram of *Cryptobranchus* femur. (*f–i*) *Ornithorhynchus* femur. (*g*) Vector map of all anisotropy, (*h*) vector map of anisotropy 0.65 and above, (*i*) vector map of the middle region in the coronal plane with anisotropy 0.65 and above. (*j*) Anisotropy frequency histogram of *Ornithorhynchus* femur. (*k–n*) *Amblyrhynchus* femur. (*l*) Vector map of all anisotropy, (*m*) vector map of anisotropy of 0.65 and above, (*n*) vector map of the mid-region of the bone in the coronal plane with anisotropy 0.65 or higher. (*o*) Anisotropy frequency histogram of *Amblyrhynchus* femur. Scale bars represent 10 mm.

tetrapods at this time) and our comparative sample is exclusively secondarily aquatic vertebrates with the exception of one *Eusthenopteron* humerus. The *Eusthenopteron* humerus MHNM 06-2674A was used to gauge the trabecular and cortical structure of a primarily aquatic vertebrate; however, drawing conclusions from comparisons of non-homologous limb elements may be misleading. Further work expanding the sample size of aquatic extant and extinct femoral elements for comparison with Blue Beach material is required to more confidently discriminate between patterns of locomotion the animals engaged in.

Differences in the compactness graphs may indicate subtle variations in aquatic locomotion such as kicking, bottom crawling, paddling or perhaps the primary or secondary nature of the aquatic limb. It may also reflect a phylogenetic signal between the Blue Beach femora or a deeper fundamental physiological or developmental difference between extinct and extant animals.

Patterns in the alignment of regions with high anisotropy were clearly visible as we predicted in extant terrestrial animals, especially at the proximal and distal ends (figure 2). In *Eublepharis*, the orientation of the highly anisotropic regions is more consistent with the transmittance of force along the bone proximodistally; however, in *Uromastyx*, the orientation of the highly anisotropic regions appears to be at a 45° angle to the proximodistal axis of the bone. We interpret this to indicate a

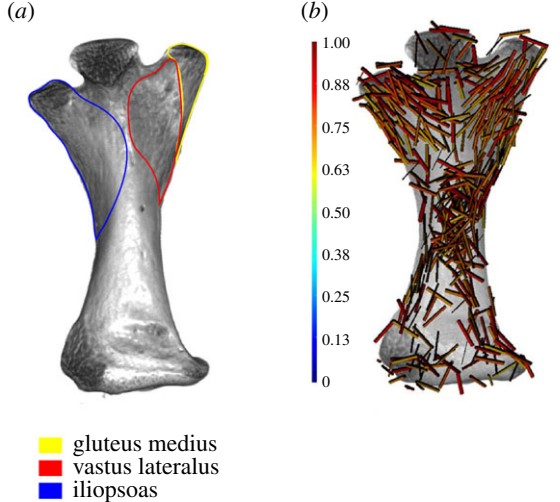

**Figure 4.** *Ornithorhynchus* femur. (*a*) Muscle map adapted from Gambaryan *et al.* [62]. Only the muscles corresponding to the aligned anisotropy vectors are illustrated here. (*b*) Vector map of the high anisotropy vectors (0.65 and above) of the whole femur.

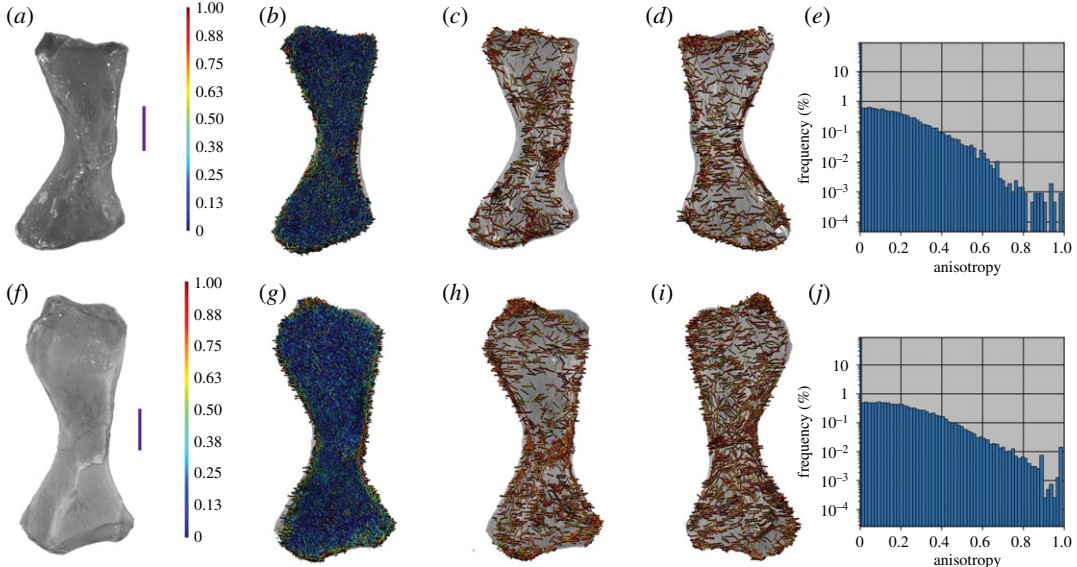

**Figure 5.** Anisotropy vector maps and histograms of fossil femora. (*a–d*) NSM.005.GF.045.043 embolomerous femur. (*b*) Vector map of all anisotropy, (*c*) vector map of anisotropy 0.65 and above in the flexor half of the bone, (*d*) vector map of anisotropy 0.65 and above in the extensor half of the bone. (*e*) Anisotropy frequency histogram of NSM.005.GF.045.043. (*f–i*) NSM.007.GF.004.630 indet. femur. (*g*) Vector map of all anisotropy, (*h*) vector map of anisotropy 0.65 and above in the flexor half of the bone, (*i*) vector map of anisotropy 0.65 and above in the extensor half of the bone. (*j*) Anisotropy frequency histogram of NSM.007.GF.004.630. Scale bars represent 10 mm.

muscle attachment site rather than remodelling due to gravitational and ground reaction forces associated with walking in a terrestrial environment. Semi-aquatic animals lacked highly anisotropic areas in the proximal and distal regions (figure 3), and only the platypus *Ornithorhynchus* showed a large, patterned region of anisotropy (figure 4). This anisotropy is superficial, at the periphery of the trabecular network, and it matches better with musculature acting on the region (gluteus medius, vastus lateralis and iliopsoas) [62] than gravitational forces. The marine iguana *Amblyrhynchus* also had a small region of superficial patterned trabecular bone (figure 3) that matches the attachment site of the puboischiofemoralis internus of *Iguana* [3], suggesting that vector maps of limb bones potentially may be informative of the presence of musculature when surface features do not provide obvious osteological correlates. The highly anisotropic areas of the fossil femora are located exclusively along the periphery of the trabecular structure (i.e. where the delineation between cortical

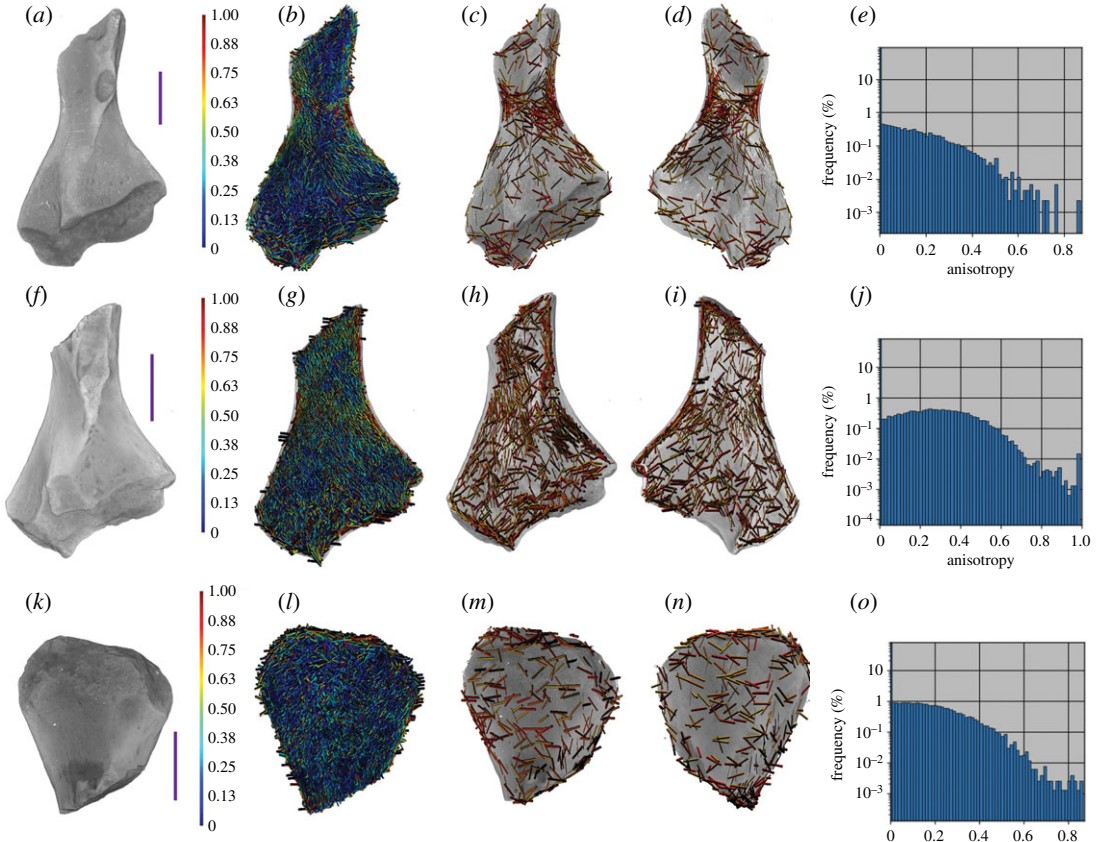

**Figure 6.** Anisotropy vector maps and historgrams of partial fossil femora. (*a–d*) NSM.005.GF.045.044 *Ossinodus*-like distal femur. (*b*) Vector map of all anisotropy, (*c*) vector map of anisotropy 0.65 and above in the flexor half of the bone, (*d*) vector map of anisotropy 0.65 and above in the extensor half of the bone. (*e*) Anisotropy frequency histogram of NSM.005.GF.045.044. (*f–i*) NSM.005.GF.045.047 whatcheerid distal femur. (*g*) Vector map of all anisotropy, (*h*) vector map of anisotropy 0.65 and above in the flexor half of the bone, (*i*) vector map of anisotropy 0.65 and above in the extensor half of the bone. (*j*) Anisotropy frequency histogram of NSM.005.GF.045.047. (*k–n*) NSM.005.GF.045.047 whatcheerid proximal femur. (*l*) Vector map of all anisotropy, (*m*) vector map of anisotropy 0.65 and above in the flexor half of the bone anisotropy 0.65 and above in the flexor half of the bone, (*n*) vector map of anisotropy 0.65 and above in the extensor half of the bone. (*o*) Anisotropy frequency histogram of NSM.005.GF.045.047. Scale bars represent 10 mm.

bone and trabecular bone occurred during segmentation; figures 5 and 6) and did not appear patterned as in *Ornithorhynchus* (figure 4). This is probably a factor of segmentation and does not convey a biological signal in terms of orientation of the trabecular structure, but it indicates that the trabecular bone thickens in such a way that, as it approaches the cortical bone, the two layers were difficult to segment from each other. This again makes the fossil femora dissimilar to both extant semi-aquatic and terrestrial animals. Previous studies which focused on analysing trabecular anisotropy and BV/ TV from small VOIs present inferences about lifestyle (i.e. fossorial, aquatic, semi-aquatic, climbing, leaping, quadrupedal, bipedal), whereas we are simply asking if alignment of the trabecular bone is consistent with remodelling in response to gravitational forces.

The lack of patterned alignment to trabecular anisotropy vectors of the Blue Beach femora, especially at the proximal and distal ends (figures 5 and 6), is unlike any of the extant animals samples, and most similar to the humerus of the primarily aquatic tetrapodomorph fish *Eusthenopteron*. This may indicate that the Blue Beach fossil femora are more similar to other fossil femora, or perhaps more similar to primarily aquatic limb bones; however, it must be noted that the *Eusthenopteron* bone is that of a humerus. Constraining the comparative sample to femora, the Blue Beach samples appear more consistent with extant semi-aquatic animals (figure 3) than terrestrial animals (figure 2). None of the fossil femora had clear alignment of high anisotropy vectors at potential muscle attachment points (shaft, or adductor blades and crests) or at joint articulation surfaces, suggesting there are not enough muscular or terrestrial forces acting on the bones to induce remodelling, but joint articulation surfaces are often worn by taphonomic forces.

**Table 2.** Patterning and frequency of anisotropy in trabecular bone of Blue Beach femora and extant femora of known locomotory behaviour.

| specimen | aligned regions of high anisotropy | most frequent anisotropy values |
|---|---|---|
| **Blue Beach limb bones** | | |
| NSM.005.GF.045.043 | no | <0.5 |
| NSM.007.GF.004.630 | no | <0.5 |
| NSM.005.GF.045.044 | no | <0.5 |
| NSM.005.GF.045.047 | no | <0.5 |
| **aquatic femora** | | |
| *Eusthenopteron* (humerus) MHNM 06-2674A | no | <0.5 |
| **semi-aquatic femora** | | |
| *Amblyrhynchus* FMNH UF41558 | weakly (superficial) at muscle insertion | <0.5 |
| *Ornithorhynchus* MVZ:32885 | weakly (superficial) at muscle insertion | <0.5 |
| **terrestrial femora** | | |
| *Eublepharis* | yes | consistent frequency across values |
| *Felis* | yes | >0.5 |
| *Uromastyx* | yes | >0.5 |

The frequency of anisotropy histograms may be an informative piece of data for future analyses linking trabecular structure to inferred lifestyle. Two of the three extant terrestrial femora show peak frequencies of anisotropy above a value of 0.4, suggesting a pattern of a greater number of anisotropic as opposed to isotropic regions of trabecular structure within the whole bone. Conversely, the semi-aquatic taxa have more frequent values of anisotropy below 0.4; therefore, across the whole bone, the trabecular structure is more isotropic than anisotropic.

Principal component (PC) analyses suggest that the Blue Beach fossil femora vary in a way that is more similar to modern aquatic animals and not modern terrestrial or semi-aquatic animals. PC1 varies most in volume and area measurements (figure 7a). These are probably due to the size of elements and not associated with locomotion. PC2 (figure 7b) varies most in cortical thickness, trabecular thickness and cortical area, probably signalling variance in bone deposition, possibly related to locomotion; however, PC2 alone does not delineate well between any of the locomotor strategies, aside from the observation that semi-aquatic animals appear to vary the most and terrestrial animals the least in the measurements that influence this component. PC3 (figure 7c) varies the most in global observed compactness, location of change in compactness and mode of anisotropy. There appears to be signalling within the variation of overall compactness of the bones in association with trabecular bone location and orientation. The axis PC3 has overlap between aquatic, semi-aquatic and terrestrial; however, there does appear to be a gradual transition from aquatic to semi-aquatic, to terrestrial animals along the PC3 axis.

The fourth PC (figure 7a–c) is heavily influenced by $R_{min}$, the compactness at the centre of the midshaft. This is supported by the findings of Quemeneur *et al.* [8] that $R_{min}$ is the most important variable for predicting aquatic or terrestrial lifestyle from limb bones. All of PC1, PC2 and PC3, when plotted against PC4, show an overlap between the Blue Beach group and aquatic group, and a separate terrestrial group of animals. The axis that is component 4 clearly delineates between aquatic, semi-aquatic and terrestrial extant femora, with the *Tulerpeton*-like Blue Beach femur NSM.007.GF.004.630 overlapping slightly in the PC4 axis with the terrestrial *Eublepharis*. The distal end of the partial *Ossinodus*-like femur NSM.005.GF.045.047-D from Blue Beach overlaps in the PC4 axis with the semi-aquatic grouping; however, this may be attributed to the incomplete nature of the element. When PC4 is plotted against PC1, PC2 and PC3, the tetrapod fossil group always overlaps on both axes with the aquatic group and does not overlap in both axes with the terrestrial or semi-aquatic group.

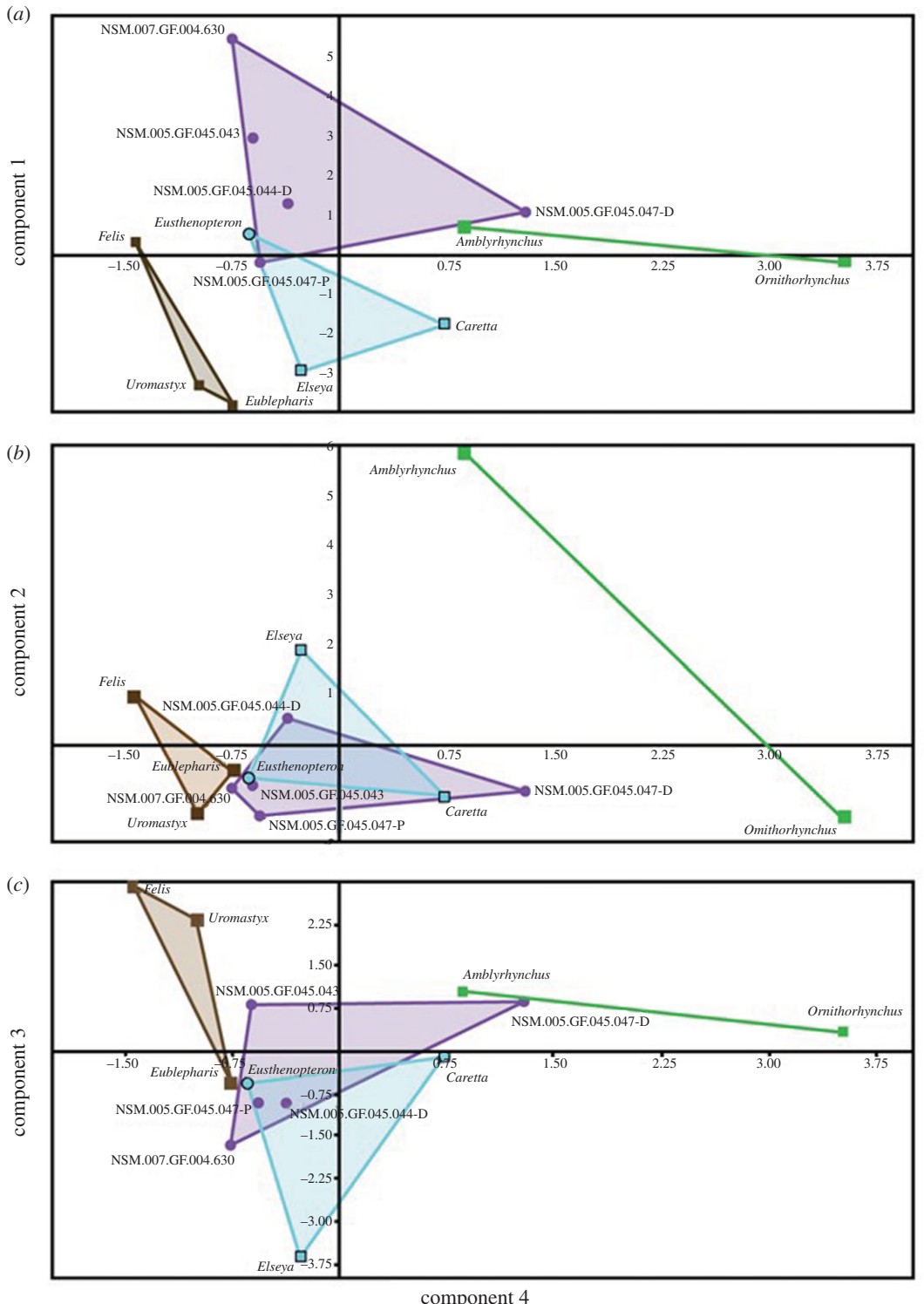

**Figure 7.** PC1, 2 and 3, by PC4. In each of these analyses, the group representing the Blue Beach femora (purple) overlap with the group representing the modern aquatic femora and *Eusthenopteron* (blue), but not the semi-aquatic (green) or terrestrial femora (brown). Square data points represent extant elements and circles represent fossil data points.

## 6. Conclusion

In extant femora, external morphology does not completely illustrate where soft tissues attached. Bryant & Seymour [25] observed in the living animals *Canus* and *Ursus* that partial muscle attachment can be inferred from osteological correlates 83% and 77% of the time, respectively; however, complete

sites of muscle attachment can be inferred only 34% and 26% of the time. In the absence of osteological correlates, scanning electron microscopy (Whitebone *et al*. in review [63]) is required to detect the previous presence of soft tissues. Even with advanced imaging, information indicative of the size or orientation of the attaching soft tissue is still difficult to determine [25,64]. In situations such as that of stem tetrapods, bracketed by such morphological disparity as seen between lungfish and living tetrapods, having other sources of information for function than inferences from soft tissue models will be useful to further constrain hypothetical prehistoric vertebrate behaviour. Here, we show that data derived from internal bone microstructure is more directly informative of past behaviour.

This new whole bone method of trabecular analysis has much potential, but subsequent studies are needed to refine the method. Other potentially informative quantitative measures including moments of inertia and bone volume fraction (measurements that are possible using Dragonfly ORS) will provide more data and expand the number of elements that are useful for examination since external morphological features, should they be present for a given soft tissue attachment, are subject to wear, erosion and inconsistent interpretation. Effective three-dimensional modelling requires rare, complete specimens for maximal informativeness.

What is clear from the fossils at Blue Beach is that there was a variety of different animals living in an aquatic environment during the early Tournaisian. These results are consistent with a long fuse between the acquisition of relatively well-developed limbs and emergence of terrestrial locomotion. Our findings are consistent with those of recent analyses concluding that early tetrapod forelimbs went through a phase of limb–substrate interaction [20] and that the humerus was well adapted to a walking gait rather than swimming [6]. We, however, provide further detail about the environment within which these limb motions were made. The pattern of anisotropy in the femora of Blue Beach tetrapods suggests that these animals were not traversing a terrestrial landscape. Given these results, we suggest that the known Blue Beach tetrapods from which body fossils are known probably 'walked' in aquatic environments.

The presence of trackways at Blue Beach presents a conundrum, given these results. It is probable that remnants of the trackway makers have not yet been found within the accumulated sample of body fossils at Blue Beach. Alternatively, it is possible that the trackways were made in an aquatic environment and subsequently exposed to air. Trackways are difficult to interpret and the sample at Blue Beach requires more thorough analysis.

This conundrum extends to trackways attributed to terrestrial tetrapods in the Devonian in Poland [65], which long predate the first body fossils that appear to belong to obligatorily aquatic animals. It has been suggested that these first tetrapod body fossils represent secondarily aquatic taxa, and that terrestrial limbs might have originated earlier in the Devonian, as suggested by the presence of trackways [2]. Using our new whole bone anisotropy analysis on a wider sample of older and younger taxa will provide a better constraint on which taxa first walked on land.

The presence of aquatic taxa represented by body fossils at Blue Beach suggests there was a diverse faunal community in an aquatic environment. The driving biotic and abiotic forces behind this aquatic diversification remain unclear, as does what prompted a water-to-land transition.

Data accessibility. Additional data are available in electronic supplementary material.

Authors' contributions. This study was conceived and designed by K.I.L. and J.S.A. Preparation of and access to specimens as well as review of the manuscript were provided by C.F.M. K.I.L. provided additional preparation, analysis and interpretation of data. S.L.M. connected K.I.L. with Dragonfly ORS and contributed to interpretation of the data. The manuscript was written by K.I.L. with assistance from J.S.A.

Competing interests. We declare we have no competing interests.

Funding. This study was supported by a Discovery Grant from the Natural Sciences and Engineering Research Council of Canada to J.S.A.

Acknowledgements. We would like to thank the Nova Scotia Museum, in particular Katherine Ogden, as well as Anthony Howell (RM), Daniel Brinkman (YPM) and Johanne Kerr (MHNM) for access to fossil specimens. We thank Ramon Nagesan for turtle scan suggestions. We would also like to thank Steven Boyd and Ying Zhu for access to and training on the Xradia scanner at the University of Calgary, and Natalia Reznikov at Dragonfly for training and assistance with the Bone Analysis function of Dragonfly. Finally, we would like to thank the two anonymous reviewers and editors for improving the manuscript.

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
