## [Peer Review File · Royal Society Open Science]

Review History

RSOS-210281.R0 (Original submission)

Review form: Reviewer 1 (Damien Germain)

Is the manuscript scientifically sound in its present form?

Yes

Are the interpretations and conclusions justified by the results?

Yes

Is the language acceptable?

Yes

Do you have any ethical concerns with this paper?

No

Have you any concerns about statistical analyses in this paper?

Yes

Recommendation?

Accept with minor revision (please list in comments)

Comments to the Author(s)

Dear authors,

I really want to congratulate you for this study.

Although it is now well known that the first tetrapods were aquatic, we still see too many studies considering the appearance of limb bones correlated with a terrestrial lifestyle. Your work is a great step toward a better comprehension of the lifestyle in the first tetrapods.

I have just a few minor comments and I corrected some minor typos in the pdf file (see Appendix A).

The only concern about statistical analyses is about the principal component analysis. It is usually better to use phylogenetic PCA in order to correct phylogenetic effects but in this case, it will be difficult to use because of the uncertainty of the phylogenetic position of the fossil specimens. I just suggest you to explain why you did not use phylogenetic PCA.

I am especially grateful to the authors for their supplemental information concerning the step-by-step fossil bone segmentation method. This is really important to allow replication and still too rare in publications.

It would be good to cite the old literature about bone functional adaptation, first described by Roux (1885) and the 'Wolff law' (Wolff 1892).

Roux, Wilhem. (1885). Beitrage zur Morphologie der funktionellen Anpassung. Archives of Anatomy and Physiology, 9, 120-185.

Wolff, Julius. (1892). Das Gesetz der Transformation der Knochen. Berlin : A. Hirchwild.

p.8 l.135. You write that terrestrial animals have thicker layers of cortical bone but this is not totally true. In small terrestrial species, cortical bone tends to be thin whereas it tends to be thicker in heavy ones or in amphibious and aquatic taxa with a passive buoyancy control.

p.12 l.231. It may be phylogenetic too

Finally, the method you developed will be very useful to assess the lifestyle of extinct taxa, not only for the first tetrapods but also many other taxa. I am sure this article will be highly cited in the future.

Review form: Reviewer 2 (Megan Whitney)

Is the manuscript scientifically sound in its present form?

No

Are the interpretations and conclusions justified by the results?

No

Is the language acceptable?

Yes

Do you have any ethical concerns with this paper?

No

Have you any concerns about statistical analyses in this paper?

Yes

Recommendation?

Major revision is needed (please make suggestions in comments)

Comments to the Author(s)

Lennie et al. investigates trabecular and cortical thickness as potential indicators of terrestriality. The authors use a modern sample with known habitat preferences and compare these modern analyses to fossils from Blue Beach, Nova Scotia. These fossils represent some of the earliest tetrapods and as such represent important data points in our understanding of vertebrate terrestrialization. The authors do establish the importance of finding more direct evidence of terrestriality than previous methods and include a very important dataset of fossils. I do believe these are important data and should be published, however I think there are substantial changes to the manuscript, interpretations, and perhaps even the sampling that are required prior to accepting this manuscript.

I have made some more specific comments in my marked version of the manuscript (see Appendix B) and I include some general themes from my comments here.

In the introduction to microstructure as a proxy for habitat (i.e. aquatic vs. terrestrial), there are two important points missing. First, I think it is important that the authors discuss the variability in aquatic indicators from modern taxa. There really is no 'smoking gun' in bone microstructure and the modern approach is to tackle this question looking a suite of features. This study does just that which I think is important, but I would urge more caution when describing what boney signals indicate. See discussion in Plasse et al. 2019 and Houssaye et al. 2016. Second, I would recommend including more sources describing experimental data on the effects of loading on trabeculae and cortical thickness. This would bolster the support for these features as possible indicators for terrestriality.

The authors refer to their methodology as novel, however, I am not entirely clear how it is new. Is it the more holistic approach to trabecular anisotropy? The novel aspect of this method needs to be more clearly stated and if it is indeed novel, it requires in-depth descriptions of the method.

There are many issues with the extant sample used in this study. It is unclear why all of the extant specimen with CT data were not run through all of the analyses. For instance, why are only the platypus and cat used in cortical thickness comparisons? There is no consistency between the extant sample included in the writing, the figures, and the table. Furthermore, the designation of aquatic vs. semi-aquatic is not consistent and the inclusion of semi-aquatic. I would recommend increasing the sample especially since there are semi-aquatic taxa included.

The inclusion of Eusthenopteron is interesting, however, it is never discussed in the text.

The conclusion would be more robust if there were statistical analyses, even simple ones, to confirm their results. To me, there are some interpretations that I don't entirely agree with, but it is hard to say for sure since as presented, the interpretations are a matter of judgement.

Finally, I think caution in grouping all of these fossils together should be a bit more prevalent in the conclusions. The authors do acknowledge the fact that there were likely many contemporaneous levels of terrestriality, and I think they're results are actually strong on that point. To me, it appears that there are some subtle differences, but as stated above, it is difficult to say without statistics. It is more likely than not that these animals were doing a variety of actions with their limbs and so I would caution making one designation for all early tetrapods based on these data.

Review form: Reviewer 3 (Dorota Konietzko-Meier)

Is the manuscript scientifically sound in its present form?

Yes

Are the interpretations and conclusions justified by the results?

Yes

Is the language acceptable?

Yes

Do you have any ethical concerns with this paper?

No

Have you any concerns about statistical analyses in this paper?

No

Recommendation?

Accept with minor revision (please list in comments)

Comments to the Author(s)

LOCOMOTORY BEHAVIOUR OF EARLY TETRAPODS FROM BLUE BEACH, NOVA SCOTIA REVEALED BY NOVEL MICROANATOMICAL ANALYSIS

Dear Editor of Royal Society Open Science

Dear Authors,

I did a review of the manuscript entitled "LOCOMOTORY BEHAVIOUR OF EARLY TETRAPODS FROM BLUE BEACH, NOVA SCOTIA REVEALED BY NOVEL MICROANATOMICAL ANALYSIS" by Kendra Lennie, Sarah Manske, Chris F Mansky. and Jason Anderson.

It is a very interesting and important contribution about the paleobiology of the early tetrapods from Devonian of Nova Scotia. It is a very welcome voice which increase our understanding of the transition from water to land, the problem which is still not fully clear. Moreover, authors are using an absolutely new method for paleontology, namely analysis of the orientation of the trabeculae. It is a very promising, non-invasive idea, complementary to early known method of the analysis of the 2D microanatomy, which, according, to the preliminary results really can help to work on the limited material without destroying it.

Although the issue is really interesting and deserves to be published in Royal Society Open Science, I have still found few problems which should be assessed before final acceptance of the paper.

1. The first is a structure of the paper. As I have checked others paleo-papers published in the journal, usually they have a standard structure with four main parts: introduction, material and methods, results and discussion. In your paper there is no clear material and method chapter - you have background which somehow provide information typical for M&M part, but in my opinion should be a little bit more structured. Your results part includes also few interpretations, which in standard structure should be in discussion.

2. The second important issue is terminology - you are using a term internal (trabecular) bone microanatomy - it is not a common definition used in paleohistology. For me it is not clear what do you mean. In mammal's histology, in long bones usually trabecular bone is an internal bone (sensu inside the bone). In early tetrapods trabecular bone it is a bone which fills the medullary region and usually is secondary, but also may build the periosteal domain (cortex, and

here could be secondary or primary). Moreover, the border between medullary region and cortex is not sharp – mostly we have there wide perimedullary region. Doing an analysis in your study of the internal trabecular microstructure you mean only bone which fills medullary region or the entire cross section (2D – it is what bone profiler is doing) or bone (3d). If only inside, where is the border between periosteal domain and trabecular bone. You need to explain these details.

Moreover it will be easier for readers to follow your results and conclusions if in the introduction you will provide the basic information how the bone microstructure is connected with the mode of life (how can you deduce about mode of life based on the bone microstructure?).

For more information about paleohistological terminology you can check:

Francillon-Vieillot, H., V. de Buffrénil, J. Castanet, J. Géraudie, F.J. Meunier, J.Y. Sire, L. Zylberberg, and A. de Ricqlès. 1990. Microstructure and mineralization of vertebrate skeletal tissues; pp. 471–530 in J.G. Carter (ed.), *Skeletal Biomineralization: Patterns, Processes and Evolutionary Trends*, Vol. I. Van Nostrand Reinhold, New York.

3. The sectioning planes for microanatomy in 2d – it is important for partially preserved bones. For early tetrapods the midshaft (histological midshaft) is very short and even a small shifting of the sectioning plane in the proximal or distal direction may results in different microanatomical picture (and then different interpretation, mostly in favor of aquatic environment). I think you can discuss the influence of the sectioning plane on your results a little bit more (you did this for one bone very short, but others I am not sure if you are in midshaft).

4. References – in some point your statements are too arbitral. You need more references to support given opinion. Moreover, I think that useful references for you will be:

Houssaye, A., Sander, M. P., and Klein, N. (2016). Adaptive patterns in aquatic amniote bone microanatomy - More complex than previously thought. *Integrative and Comparative Biology* 2016:1–21. DOI:10.1093/icb/icw120.

Sanchez, S., Tafforeau, P., Ahlberg, P. (2014). The humerus of Eusthenopteron: a puzzling organization presaging the establishment of tetrapod limb bone marrow. *Proceedings of the Royal Society of London. Biological Sciences*, ISSN 0962-8452, E-ISSN 1471-2954, Vol. 281, no 1782, p. 20140299

Sanchez, S., Dupret, V., Tafforeau, P., Trinajstić, K., Ryll, B. et al. (2013). 3D Microstructural Architecture of Muscle Attachments in Extant and Fossil Vertebrates Revealed by Synchrotron Microtomography. *SPLoS ONE*, vol. 8, ss. e56992- DOI

Sanchez, S., Tafforeau, P., Clack, J., Ahlberg, P. (2016). Life history of the stem tetrapod *Acanthostega* revealed by synchrotron microtomography. *Nature*, vol. 537, ss. 408-+ DOI

Qu, Q., Sanchez, S., Zhu, M., Blom, H., Ahlberg, P. (2017). The origin of novel features by changes in developmental mechanisms: ontogeny and three-dimensional microanatomy of polyodontode scales of two early osteichthyans. *Biological Reviews*, vol. 92, ss. 1189-1212.

Especially the first is important, as authors in that paper show that conclusions about mode of life only based on the 2d microanatomy are not always very easy. Thus, your method of the analysis of the “trabecular” orientation is very important as a supporting source of information. Other papers are not dealing directly with the same topic as yours, but I think in introduction it will be nice to have short overview what was done up to date, as few attempts to use 3d modelling for the life reconstruction of early “tetrapods” are also known.

5. It will be nice to have a scale bars in your figures – I think that you have a scale bars, but there is no information what does it mean...

Finally, I would like to recommend the paper for publication after moderate revision. Please see attached pdf (Appendix C) with my comments. Should the authors have any questions regarding my comments, they are welcome to contact me.

Decision letter (RSOS-210281.R0)

Dear Ms Lennie

The Editors assigned to your paper RSOS-210281 "Locomotory behaviour of early tetrapods from Blue Beach, Nova Scotia revealed by novel microanatomical analysis" have now received comments from reviewers and would like you to revise the paper in accordance with the reviewer comments and any comments from the Editors. Please note this decision does not guarantee eventual acceptance.

A number of attachments have been included with this email, please let the editorial office know if these are not received.

Please submit your revised manuscript and required files (see below) no later than 21 days from today's (ie 23-Mar-2021) date. Note: the ScholarOne system will 'lock' if submission of the revision is attempted 21 or more days after the deadline. If you do not think you will be able to meet this deadline please contact the editorial office immediately.

on behalf of Dr Jennifer Botha (Associate Editor) and Kevin Padian (Subject Editor)
 openscience@royalsociety.org

Associate Editor Comments to Author (Dr Jennifer Botha):

Associate Editor: 1

Comments to the Author:

Dear authors,

Two reviews have been received regarding your article. One reviewer recommended minor revision. However, the second reviewer recommended major revision with several recommendations for strengthening the manuscript. Many queries require comments particularly those regarding the sampling and statistical analysis. Several comments have been annotated on the manuscript itself (from both reviewers).

Reviewer comments to Author:

Reviewer: 1

Comments to the Author(s)

Dear authors,

I really want to congratulate you for this study.

Although it is now well known that the first tetrapods were aquatic, we still see too many studies considering the appearance of limb bones correlated with a terrestrial lifestyle. Your work is a great step toward a better comprehension of the lifestyle in the first tetrapods.

I have just a few minor comments and I corrected some minor typos in the pdf file.

The only concern about statistical analyses is about the principal component analysis. It is usually better to use phylogenetic PCA in order to correct phylogenetic effects but in this case, it will be difficult to use because of the uncertainty of the phylogenetic position of the fossil specimens. I just suggest you to explain why you did not use phylogenetic PCA.

I am especially grateful to the authors for their supplemental information concerning the step-by-step fossil bone segmentation method. This is really important to allow replication and still too rare in publications.

It would be good to cite the old literature about bone functional adaptation, first described by Roux (1885) and the 'Wolff law' (Wolff 1892).

Roux, Wilhem. (1885). Beitrage zur Morphologie der funktionellen Anpassung. Archives of Anatomy and Physiology, 9, 120-185.

Wolff, Julius. (1892). Das Gesetz der Transformation der Knochen. Berlin : A. Hirschwild.

p.8 l.135. You write that terrestrial animals have thicker layers of cortical bone but this is not totally true. In small terrestrial species, cortical bone tends to be thin whereas it tends to be thicker in heavy ones or in amphibious and aquatic taxa with a passive buoyancy control.

p.12 l.231. It may be phylogenetic too

Finally, the method you developed will be very useful to assess the lifestyle of extinct taxa, not only for the first tetrapods but also many other taxa. I am sure this article will be highly cited in the future.

Reviewer: 2

Comments to the Author(s)

Lennie et al. investigates trabecular and cortical thickness as potential indicators of terrestriality. The authors use a modern sample with known habitat preferences and compare these modern analyses to fossils from Blue Beach, Nova Scotia. These fossils represent some of the earliest tetrapods and as such represent important data points in our understanding of vertebrate terrestrialization. The authors do establish the importance of finding more direct evidence of terrestriality than previous methods and include a very important dataset of fossils. I do believe these are important data and should be published, however I think there are substantial changes to the manuscript, interpretations, and perhaps even the sampling that are required prior to accepting this manuscript.

I have made some more specific comments in my marked version of the manuscript and I include some general themes from my comments here.

In the introduction to microstructure as a proxy for habitat (i.e. aquatic vs. terrestrial), there are two important points missing. First, I think it is important that the authors discuss the variability in aquatic indicators from modern taxa. There really is no 'smoking gun' in bone microstructure and the modern approach is to tackle this question looking a suite of features. This study does just that which I think is important, but I would urge more caution when describing what boney signals indicate. See discussion in Plasse et al. 2019 and Houssaye et al. 2016. Second, I would recommend including more sources describing experimental data on the effects of loading on trabeculae and cortical thickness. This would bolster the support for these features as possible indicators for terrestriality.

The authors refer to their methodology as novel, however, I am not entirely clear how it is new. Is it the more holistic approach to trabecular anisotropy? The novel aspect of this method needs to be more clearly stated and if it is indeed novel, it requires in-depth descriptions of the method.

There are many issues with the extant sample used in this study. It is unclear why all of the extant specimen with CT data were not run through all of the analyses. For instance, why are only the platypus and cat used in cortical thickness comparisons? There is no consistency between the extant sample included in the writing, the figures, and the table. Furthermore, the designation of aquatic vs. semi-aquatic is not consistent and the inclusion of semi-aquatic. I would recommend increasing the sample especially since there are semi-aquatic taxa included.

The inclusion of Eusthenopteron is interesting, however, it is never discussed in the text.

The conclusion would be more robust if there were statistical analyses, even simple ones, to confirm their results. To me, there are some interpretations that I don't entirely agree with, but it is hard to say for sure since as presented, the interpretations are a matter of judgement.

Finally, I think caution in grouping all of these fossils together should be a bit more prevalent in the conclusions. The authors do acknowledge the fact that there were likely many contemporaneous levels of terrestriality, and I think they're results are actually strong on that point. To me, it appears that there are some subtle differences, but as stated above, it is difficult to say without statistics. It is more likely than not that these animals were doing a variety of actions with their limbs and so I would caution making one designation for all early tetrapods based on these data.

===PREPARING YOUR MANUSCRIPT===

===PREPARING YOUR REVISION IN SCHOLARONE===

- An editable file of each table (.doc, .docx, .xls, .xlsx, or .csv).
- An editable file of all figure and table captions.

- Any electronic supplementary material (ESM).
- If you are requesting a discretionary waiver for the article processing charge, the waiver form must be included at this step.
- If you are providing image files for potential cover images, please upload these at this step, and inform the editorial office you have done so. You must hold the copyright to any image provided.
- A copy of your point-by-point response to referees and Editors. This will expedite the preparation of your proof.

- Ensure that your data access statement meets the requirements at <https://royalsociety.org/journals/authors/author-guidelines/#data>. You should ensure that you cite the dataset in your reference list. If you have deposited data etc in the Dryad repository, please include both the 'For publication' link and 'For review' link at this stage.
- If you are requesting an article processing charge waiver, you must select the relevant waiver option (if requesting a discretionary waiver, the form should have been uploaded at Step 3 'File upload' above).
- If you have uploaded ESM files, please ensure you follow the guidance at <https://royalsociety.org/journals/authors/author-guidelines/#supplementary-material> to include a suitable title and informative caption. An example of appropriate titling and captioning may be found at https://figshare.com/articles/Table_S2_from_Is_there_a_trade-off_between_peak_performance_and_performance_breadth_across_temperatures_for_aerobic_scope_in_teleost_fishes_/3843624.

Author's Response to Decision Letter for (RSOS-210281.R0)

See Appendices D & E.

Decision letter (RSOS-210281.R1)

Dear Ms Lennie,

It is a pleasure to accept your manuscript entitled "Locomotory behaviour of early tetrapods from Blue Beach, Nova Scotia revealed by novel microanatomical analysis" in its current form for publication in Royal Society Open Science. The comments of the reviewer(s) who reviewed your manuscript are included at the foot of this letter.

You can expect to receive a proof of your article in the near future. Please contact the editorial office (openscience@royalsociety.org) and the production office (openscience_proofs@royalsociety.org) to let us know if you are likely to be away from e-mail contact – if you are going to be away, please nominate a co-author (if available) to manage the proofing process, and ensure they are copied into your email to the journal.

on behalf of Dr Jennifer Botha (Associate Editor) and Kevin Padian (Subject Editor)
openscience@royalsociety.org

Associate Editor Comments to Author (Dr Jennifer Botha):

Comments to the Author:

Thank you for satisfactorily answering the reviewers queries and adding in the requested information.

Appendix A**ROYAL SOCIETY
OPEN SCIENCE****Locomotory behaviour of early tetrapods from Blue Beach,
Nova Scotia revealed by novel microanatomical analysis**

Journal:	Royal Society Open Science
Manuscript ID	RSOS-210281
Article Type:	Research
Date Submitted by the Author:	18-Feb-2021
Complete List of Authors:	Lennie, Kendra; University of Calgary, Biological Sciences Manske, Sarah; University of Calgary, Radiology Mansky, Chris F.; Blue Beach Fossil Museum Anderson, Jason; University of Calgary
Subject:	palaeontology < BIOLOGY, evolution < BIOLOGY, biomechanics < BIOLOGY
Keywords:	tetrapods, Carboniferous, Locomotion, Devonian, Fin-to-limb, Water-to-land
Subject Category:	Organismal and Evolutionary Biology

Author-supplied statements

Relevant information will appear here if provided.

Ethics

Does your article include research that required ethical approval or permits?:

This article does not present research with ethical considerations

Statement (if applicable):

CUST_IF_YES_ETHICS :No data available.

Data

It is a condition of publication that data, code and materials supporting your paper are made publicly available. Does your paper present new data?:

Yes

Statement (if applicable):

additional data in supplementary materials

Conflict of interest

I/We declare we have no competing interests

Statement (if applicable):

CUST_STATE_CONFLICT :No data available.

Authors' contributions

This paper has multiple authors and our individual contributions were as below

Statement (if applicable):

This study was conceived and designed by KIL and JSA. Preparation of and access to specimens as well as review of the manuscript was provided by CFM. KIL provided additional preparation, analysis, and interpretation of data. SLM connected KIL with Dragonfly ORS and contributed to interpretation of the data. The manuscript was written by KIL with assistance from JSA.

Locomotory behaviour of early tetrapods from Blue Beach, Nova Scotia revealed by novel
microanatomical analysis

*Kendra I. Lennie^{1,2}

Sarah L. Manske^{2,3}

Chris F. Mansky⁴

Jason S. Anderson^{2,5}

University of Calgary, Biological Sciences. 507 Campus Drive N.W. University of Calgary.

Calgary, Alberta. Canada, T2N 1N4. kendra.lennie@ucalgary.ca

University of Calgary, McCaig Institute for Bone and Joint Health. Foothills Campus, University of

Calgary. 3330 Hospital Drive NW, Calgary, Alberta, Canada. T2N 4N1

University of Calgary, Radiology. Foothills Medical Centre. 1403-29th Street NW, Calgary, Alberta.

Canada T2N 2T9

Blue Beach Fossil Museum, 127 Blue Beach Road, Hantsport, Nova Scotia, B0P 1P0.

bbfossils@xplornet.com

University of Calgary, Comparative Biology and Experimental Medicine. Foothills Campus,

[revised manuscript text omitted]

Anisotropy and Bone Volume Fraction

Behaviourally moderated repetitive forces acting on bone induce remodelling [7,37]. The organization of trabeculae therefore may provide more sensitive information about forces acting on the bone than a two-dimensional midshaft analysis. It has been suggested that results pertaining to trabecular anisotropy (the non-random alignment of a three-dimensional structure) are sensitive to the location of the volume of interest (VOI) within the bone [35,53]. For this reason, we conducted a whole bone analysis of trabecular anisotropy to maximize likelihood of capturing terrestrial signal across the entire element and find potential local orientations that we would not have sampled choosing a single area a priori.

Principal Components Analysis

To analyse potential similarities and differences more rigorously between known aquatic and terrestrial taxa and the fossils from Blue Beach we conducted a principal component analysis. Our analysis used of a combination of Dragonfly (ORS; Montréal) and Bone Profiler variables.

**Results**

*Compactness profiles*

The binary images of the femoral midshafts (Figure 1) clearly show the medullary space filled
entirely with trabecular bone, suggesting a more aquatic lifestyle. To test this, we used Bone Profiler
[22] to graph and compute compactness profiles of the Blue Beach femora. Compactness graphs
(Figure 1) show that the midshafts of the Blue Beach femora are unlike the graphs of both terrestrial
(*Felis*, cat; Figure 1) and aquatic (*Ornithorhynchus*, platypus; Figure 1) taxa; however, they are more
dissimilar to the terrestrial taxa than the aquatic taxa. The aquatic taxa have trabecular bone within the
medullary cavity and throughout the transition zone between medullary cavity and cortical bone. The
Blue Beach femora have trabecular bone within the medullary cavity, and the transition from medullary
cavity to cortical bone is gradual. Additionally, the Blue Beach femora have non-zero compactness
value at the bone centre, a trait shared with *Ornithorhynchus*, indicating the presence of trabecular bone
through the entire medullary cavity. The Blue Beach femora, which were predicted to represent a range
of expected aquatic to terrestrial taxa, were all designated aquatic.

*Trabecular anisotropy*

**Patterns of trabecular structure in modern terrestrial and amphibious tetrapods**

[revised manuscript text omitted]

45 46 47 48 49 324 **Acknowledgements**

We thank Katherine Ogdon and Tim Fedak (Nova Scotia Museum) and Johanne Kerr (Musée
d’Histoire Naturelle de Miguasha) for access to the specimens and Sonja Wood (Blue Beach Fossil
Museum) for discussions and Natalie Reznikov for training KIL on the use of Bone Analysis in

Dragonfly ORS. Thank you to Ramon Nagesan for suggesting several accessible turtle specimens. This
study was supported by a Discovery Grant from the Natural Sciences and Engineering Research
Council of Canada to JSA.

**Abbreviations**

Blue Beach Fossil Museum, Hantsport, Nova Scotia, Canada (BBFM); Florida Museum of
Natural History (FMNH); Musée d'Histoire Naturelle de Miguasha, Québec, Canada (MHNM);
Museum of Vertebrate Zoology, University of California, Berkeley, USA (MVZ); Nova Scotia Museum
of Natural History, Halifax, Canada (NSM); Redpath Museum, McGill University, Montreal, Canada
(RM); University of Florida (UF); University of Michigan Museum of Zoology, Ann Arbor, Michigan,
USA (UMMZ); Yale Peabody Museum of Natural History, New Haven, Connecticut, USA (YPM);
Museum für Naturkunde Berlin, Germany (ZMB).

**Authorship**

This study was conceived and designed by KIL and JSA. Preparation of and access to
specimens as well as review of the manuscript was provided by CFM. KIL provided additional
preparation, analysis, and interpretation of data. SLM connected KIL with Dragonfly ORS and
contributed to interpretation of the data. The manuscript was written by KIL with assistance from JSA.

**Conflict of Interest**

The authors are not aware of any conflict of interest.

**Additional Information**

Supplementary information is available for this paper. Correspondence and requests for
materials should be addressed to KIL.

[revised manuscript text omitted]

*10.4138/2076*
42. Tibert NE, Scott DB. 1999 Ostracodes and Agglutinated Foraminifera as Indicators of
Paleoenvironmental Change in an Early Carboniferous Brackish Bay, Atlantic Canada. *Palaios*
**14**, 246. (doi:10.2307/3515437)
43. Lebedev OA, Coates MI. 1995 The postcranial skeleton of the Devonian tetrapod *Tulerpeton*
*curtum* Lebedev. *Zool. J. Linn. Soc.* **114**, 307–348. (doi:10.1111/j.1096-3642.1995.tb00119.x)
44. Lombard RE, Bolt JR. 1995 New Primitive Tetrapod *Whatcheeria Deltae* From Lower
Carboniferous Iowa. *Palaeontology.* **38**.
45. Bishop PJ, Walmsley CW, Phillips MJ, Quayle MR, Boisvert CA, McHenry CR. 2015 Oldest
pathology in a tetrapod bone illuminates the origin of terrestrial vertebrates. *PLoS One* **10**,
e0125723. (doi:10.1371/journal.pone.0125723)
46. Smithson TR. 1985 The morphology and relationships of the Carboniferous amphibian
*Eoherpeton watsoni* Panchen. *Zool. J. Linn. Soc.* **85**, 317–410. (doi:10.1111/j.1096-
3642.1985.tb01517.x)
47. Lanyon LE, Goodship AE, Pye CJ, MacFie JH. 1982 Mechanically adaptive bone remodelling.
*J. Biomech.* **15**, 141–154. (doi:10.1016/0021-9290(82)90246-9)
48. Carter DR, Orr TE, Fyhrie DP. 1989 Relationships between loading history and femoral
cancellous bone architecture. *J. Biomech.* **22**, 231–44. (doi:10.1016/0021-9290(89)90091-2)
49. Canoville A, Laurin M. 2009 Microanatomical diversity of the humerus and lifestyle in
lissamphibians. *Acta Zool.* **90**, 110–122. (doi:10.1111/j.1463-6395.2008.00328.x)
50. Sanchez S, Germain D, De Ricqlès A, Abourachid A, Goussard F, Tafforeau P. 2010 Limb-bone
histology of temnospondyls: Implications for understanding the diversification of
palaeoecologies and patterns of locomotion of Permo-Triassic tetrapods. *J. Evol. Biol.* **23**, 2076–
2090. (doi:10.1111/j.1420-9101.2010.02081.x)
51. Quemeneur S, de Buffrénil V, Laurin M. 2013 Microanatomy of the amniote femur and
inference of lifestyle in limbed vertebrates. *Biol. J. Linn. Soc.* **109**, 644–655.
(doi:10.1111/bij.12066)

52. Dunmore CJ, Kivell TL, Bardo A, Skinner MM. 2019 Metacarpal trabecular bone varies with
distinct hand-positions used in hominid locomotion. *J. Anat.* **235**, 45–66.
(doi:10.1111/joa.12966)

[revised manuscript text omitted]

Journal:	Royal Society Open Science
Manuscript ID	RSOS-210281
Article Type:	Research
Date Submitted by the Author:	18-Feb-2021
Complete List of Authors:	Lennie, Kendra; University of Calgary, Biological Sciences Manske, Sarah; University of Calgary, Radiology Mansky, Chris F.; Blue Beach Fossil Museum Anderson, Jason; University of Calgary
Subject:	palaeontology < BIOLOGY, evolution < BIOLOGY, biomechanics < BIOLOGY
Keywords:	tetrapods, Carboniferous, Locomotion, Devonian, Fin-to-limb, Water-to-land
Subject Category:	Organismal and Evolutionary Biology

Author-supplied statements

Relevant information will appear here if provided.

Ethics

Does your article include research that required ethical approval or permits?:

This article does not present research with ethical considerations

Statement (if applicable):

CUST_IF_YES_ETHICS :No data available.

Data

It is a condition of publication that data, code and materials supporting your paper are made publicly available. Does your paper present new data?:

Yes

Statement (if applicable):

additional data in supplementary materials

Conflict of interest

I/We declare we have no competing interests

Statement (if applicable):

CUST_STATE_CONFLICT :No data available.

Authors' contributions

This paper has multiple authors and our individual contributions were as below

Statement (if applicable):

This study was conceived and designed by KIL and JSA. Preparation of and access to specimens as well as review of the manuscript was provided by CFM. KIL provided additional preparation, analysis, and interpretation of data. SLM connected KIL with Dragonfly ORS and contributed to interpretation of the data. The manuscript was written by KIL with assistance from JSA.

Locomotory behaviour of early tetrapods from Blue Beach, Nova Scotia revealed by novel
microanatomical analysis

*Kendra I. Lennie^{1,2}

Sarah L. Manske^{2,3}

Chris F. Mansky⁴

Jason S. Anderson^{2,5}

University of Calgary, Biological Sciences. 507 Campus Drive N.W. University of Calgary.

Calgary, Alberta. Canada, T2N 1N4. kendra.lennie@ucalgary.ca

University of Calgary, McCaig Institute for Bone and Joint Health. Foothills Campus, University of

Calgary. 3330 Hospital Drive NW, Calgary, Alberta, Canada. T2N 4N1

University of Calgary, Radiology. Foothills Medical Centre. 1403-29th Street NW, Calgary, Alberta.

Canada T2N 2T9

Blue Beach Fossil Museum, 127 Blue Beach Road, Hantsport, Nova Scotia, B0P 1P0.

bbfossils@xplornet.com

University of Calgary, Comparative Biology and Experimental Medicine. Foothills Campus,

[revised manuscript text omitted]

Anisotropy and Bone Volume Fraction

Behaviourally moderated repetitive forces acting on bone induce remodelling [7,37]. The organization of trabeculae therefore may provide more sensitive information about forces acting on the bone than a two-dimensional midshaft analysis. It has been suggested that results pertaining to trabecular anisotropy (the non-random alignment of a three-dimensional structure) are sensitive to the location of the volume of interest (VOI) within the bone [35,53]. For this reason, we conducted a whole bone analysis of trabecular anisotropy to maximize likelihood of capturing terrestrial signal across the entire element and find potential local orientations that we would not have sampled choosing a single area a priori.

Principal Components Analysis

To analyse potential similarities and differences more rigorously between known aquatic and terrestrial taxa and the fossils from Blue Beach we conducted a principal component analysis. Our analysis used of a combination of Dragonfly (ORS; Montréal) and Bone Profiler variables.

**Results**

*Compactness profiles*

The binary images of the femoral midshafts (Figure 1) clearly show the medullary space filled
entirely with trabecular bone, suggesting a more aquatic lifestyle. To test this, we used Bone Profiler
[22] to graph and compute compactness profiles of the Blue Beach femora. Compactness graphs
(Figure 1) show that the midshafts of the Blue Beach femora are unlike the graphs of both terrestrial
(*Felis*, cat; Figure 1) and aquatic (*Ornithorhynchus*, platypus; Figure 1) taxa; however, they are more
dissimilar to the terrestrial taxa than the aquatic taxa. The aquatic taxa have trabecular bone within the
medullary cavity and throughout the transition zone between medullary cavity and cortical bone. The
Blue Beach femora have trabecular bone within the medullary cavity, and the transition from medullary
cavity to cortical bone is gradual. Additionally, the Blue Beach femora have non-zero compactness
value at the bone centre, a trait shared with *Ornithorhynchus*, indicating the presence of trabecular bone
through the entire medullary cavity. The Blue Beach femora, which were predicted to represent a range
of expected aquatic to terrestrial taxa, were all designated aquatic.
*Trabecular anisotropy*

**Patterns of trabecular structure in modern terrestrial and amphibious tetrapods**

[revised manuscript text omitted]

44 45 46 47 48 49 324 **Acknowledgements**

We thank Katherine Ogdon and Tim Fedak (Nova Scotia Museum) and Johanne Kerr (Musée
d’Histoire Naturelle de Miguasha) for access to the specimens and Sonja Wood (Blue Beach Fossil
Museum) for discussions and Natalie Reznikov for training KIL on the use of Bone Analysis in

Dragonfly ORS. Thank you to Ramon Nagesan for suggesting several accessible turtle specimens. This
study was supported by a Discovery Grant from the Natural Sciences and Engineering Research
Council of Canada to JSA.

**Abbreviations**

Blue Beach Fossil Museum, Hantsport, Nova Scotia, Canada (BBFM); Florida Museum of
Natural History (FMNH); Musée d'Histoire Naturelle de Miguasha, Québec, Canada (MHNM);
Museum of Vertebrate Zoology, University of California, Berkeley, USA (MVZ); Nova Scotia Museum
of Natural History, Halifax, Canada (NSM); Redpath Museum, McGill University, Montreal, Canada
(RM); University of Florida (UF); University of Michigan Museum of Zoology, Ann Arbor, Michigan,
USA (UMMZ); Yale Peabody Museum of Natural History, New Haven, Connecticut, USA (YPM);
Museum für Naturkunde Berlin, Germany (ZMB).

**Authorship**

This study was conceived and designed by KIL and JSA. Preparation of and access to
specimens as well as review of the manuscript was provided by CFM. KIL provided additional
preparation, analysis, and interpretation of data. SLM connected KIL with Dragonfly ORS and
contributed to interpretation of the data. The manuscript was written by KIL with assistance from JSA.

**Conflict of Interest**

The authors are not aware of any conflict of interest.

**Additional Information**

Supplementary information is available for this paper. Correspondence and requests for
materials should be addressed to KIL.

[revised manuscript text omitted]

*10.4138/2076*
42. Tibert NE, Scott DB. 1999 Ostracodes and Agglutinated Foraminifera as Indicators of
Paleoenvironmental Change in an Early Carboniferous Brackish Bay, Atlantic Canada. *Palaios*
**14**, 246. (doi:10.2307/3515437)
43. Lebedev OA, Coates MI. 1995 The postcranial skeleton of the Devonian tetrapod *Tulerpeton*
*curtum* Lebedev. *Zool. J. Linn. Soc.* **114**, 307–348. (doi:10.1111/j.1096-3642.1995.tb00119.x)
44. Lombard RE, Bolt JR. 1995 New Primitive Tetrapod *Whatcheeria Deltae* From Lower
Carboniferous Iowa. *Palaeontology.* **38**.
45. Bishop PJ, Walmsley CW, Phillips MJ, Quayle MR, Boisvert CA, McHenry CR. 2015 Oldest
pathology in a tetrapod bone illuminates the origin of terrestrial vertebrates. *PLoS One* **10**,
e0125723. (doi:10.1371/journal.pone.0125723)
46. Smithson TR. 1985 The morphology and relationships of the Carboniferous amphibian
*Eoherpeton watsoni* Panchen. *Zool. J. Linn. Soc.* **85**, 317–410. (doi:10.1111/j.1096-
3642.1985.tb01517.x)
47. Lanyon LE, Goodship AE, Pye CJ, MacFie JH. 1982 Mechanically adaptive bone remodelling.
*J. Biomech.* **15**, 141–154. (doi:10.1016/0021-9290(82)90246-9)
48. Carter DR, Orr TE, Fyhrie DP. 1989 Relationships between loading history and femoral
cancellous bone architecture. *J. Biomech.* **22**, 231–44. (doi:10.1016/0021-9290(89)90091-2)
49. Canoville A, Laurin M. 2009 Microanatomical diversity of the humerus and lifestyle in
lissamphibians. *Acta Zool.* **90**, 110–122. (doi:10.1111/j.1463-6395.2008.00328.x)
50. Sanchez S, Germain D, De Ricqlès A, Abourachid A, Goussard F, Tafforeau P. 2010 Limb-bone
histology of temnospondyls: Implications for understanding the diversification of
palaeoecologies and patterns of locomotion of Permo-Triassic tetrapods. *J. Evol. Biol.* **23**, 2076–
2090. (doi:10.1111/j.1420-9101.2010.02081.x)
51. Quemeneur S, de Buffrénil V, Laurin M. 2013 Microanatomy of the amniote femur and
inference of lifestyle in limbed vertebrates. *Biol. J. Linn. Soc.* **109**, 644–655.
(doi:10.1111/bij.12066)

52. Dunmore CJ, Kivell TL, Bardo A, Skinner MM. 2019 Metacarpal trabecular bone varies with
distinct hand-positions used in hominid locomotion. *J. Anat.* **235**, 45–66.
(doi:10.1111/joa.12966)

[revised manuscript text omitted]

Journal:	Royal Society Open Science
Manuscript ID	RSOS-210281
Article Type:	Research
Date Submitted by the Author:	18-Feb-2021
Complete List of Authors:	Lennie, Kendra; University of Calgary, Biological Sciences Manske, Sarah; University of Calgary, Radiology Mansky, Chris F.; Blue Beach Fossil Museum Anderson, Jason; University of Calgary
Subject:	palaeontology < BIOLOGY, evolution < BIOLOGY, biomechanics < BIOLOGY
Keywords:	tetrapods, Carboniferous, Locomotion, Devonian, Fin-to-limb, Water-to-land
Subject Category:	Organismal and Evolutionary Biology

Author-supplied statements

Relevant information will appear here if provided.

Ethics

Does your article include research that required ethical approval or permits?:

This article does not present research with ethical considerations

Statement (if applicable):

CUST_IF_YES_ETHICS :No data available.

Data

It is a condition of publication that data, code and materials supporting your paper are made publicly available. Does your paper present new data?:

Yes

Statement (if applicable):

additional data in supplementary materials

Conflict of interest

I/We declare we have no competing interests

Statement (if applicable):

CUST_STATE_CONFLICT :No data available.

Authors' contributions

This paper has multiple authors and our individual contributions were as below

Statement (if applicable):

This study was conceived and designed by KIL and JSA. Preparation of and access to specimens as well as review of the manuscript was provided by CFM. KIL provided additional preparation, analysis, and interpretation of data. SLM connected KIL with Dragonfly ORS and contributed to interpretation of the data. The manuscript was written by KIL with assistance from JSA.

Locomotory behaviour of early tetrapods from Blue Beach, Nova Scotia revealed by novel
microanatomical analysis

*Kendra I. Lennie^{1,2}

Sarah L. Manske^{2,3}

Chris F. Mansky⁴

Jason S. Anderson^{2,5}

University of Calgary, Biological Sciences. 507 Campus Drive N.W. University of Calgary.

Calgary, Alberta. Canada, T2N 1N4. kendra.lennie@ucalgary.ca

University of Calgary, McCaig Institute for Bone and Joint Health. Foothills Campus, University of

Calgary. 3330 Hospital Drive NW, Calgary, Alberta, Canada. T2N 4N1

University of Calgary, Radiology. Foothills Medical Centre. 1403-29th Street NW, Calgary, Alberta.

Canada T2N 2T9

Blue Beach Fossil Museum, 127 Blue Beach Road, Hantsport, Nova Scotia, B0P 1P0.

bbfossils@xplornet.com

University of Calgary, Comparative Biology and Experimental Medicine. Foothills Campus,

[revised manuscript text omitted]

Anisotropy and Bone Volume Fraction

Behaviourally moderated repetitive forces acting on bone induce remodelling [7,37]. The organization of trabeculae therefore may provide more sensitive information about forces acting on the bone than a two-dimensional midshaft analysis. It has been suggested that results pertaining to trabecular anisotropy (the non-random alignment of a three-dimensional structure) are sensitive to the location of the volume of interest (VOI) within the bone [35,53]. For this reason, we conducted a whole bone analysis of trabecular anisotropy to maximize likelihood of capturing terrestrial signal across the entire element and find potential local orientations that we would not have sampled choosing a single area a priori.

Principal Components Analysis

To analyse potential similarities and differences more rigorously between known aquatic and terrestrial taxa and the fossils from Blue Beach we conducted a principal component analysis. Our analysis used of a combination of Dragonfly (ORS; Montréal) and Bone Profiler variables.

**Results**

*Compactness profiles*

The binary images of the femoral midshafts (Figure 1) clearly show the medullary space filled
entirely with trabecular bone, suggesting a more aquatic lifestyle. To test this, we used Bone Profiler
[22] to graph and compute compactness profiles of the Blue Beach femora. Compactness graphs
(Figure 1) show that the midshafts of the Blue Beach femora are unlike the graphs of both terrestrial
(*Felis*, cat; Figure 1) and aquatic (*Ornithorhynchus*, platypus; Figure 1) taxa; however, they are more
dissimilar to the terrestrial taxa than the aquatic taxa. The aquatic taxa have trabecular bone within the
medullary cavity and throughout the transition zone between medullary cavity and cortical bone. The
Blue Beach femora have trabecular bone within the medullary cavity, and the transition from medullary
cavity to cortical bone is gradual. Additionally, the Blue Beach femora have non-zero compactness
value at the bone centre, a trait shared with *Ornithorhynchus*, indicating the presence of trabecular bone
through the entire medullary cavity. The Blue Beach femora, which were predicted to represent a range
of expected aquatic to terrestrial taxa, were all designated aquatic.

*Trabecular anisotropy*

**Patterns of trabecular structure in modern terrestrial and amphibious tetrapods**

[revised manuscript text omitted]

44 45 46 47 48 49 324 **Acknowledgements**

We thank Katherine Ogdon and Tim Fedak (Nova Scotia Museum) and Johanne Kerr (Musée
d’Histoire Naturelle de Miguasha) for access to the specimens and Sonja Wood (Blue Beach Fossil
Museum) for discussions and Natalie Reznikov for training KIL on the use of Bone Analysis in

Dragonfly ORS. Thank you to Ramon Nagesan for suggesting several accessible turtle specimens. This
study was supported by a Discovery Grant from the Natural Sciences and Engineering Research
Council of Canada to JSA.

**Abbreviations**

Blue Beach Fossil Museum, Hantsport, Nova Scotia, Canada (BBFM); Florida Museum of
Natural History (FMNH); Musée d'Histoire Naturelle de Miguasha, Québec, Canada (MHNM);
Museum of Vertebrate Zoology, University of California, Berkeley, USA (MVZ); Nova Scotia Museum
of Natural History, Halifax, Canada (NSM); Redpath Museum, McGill University, Montreal, Canada
(RM); University of Florida (UF); University of Michigan Museum of Zoology, Ann Arbor, Michigan,
USA (UMMZ); Yale Peabody Museum of Natural History, New Haven, Connecticut, USA (YPM);
Museum für Naturkunde Berlin, Germany (ZMB).

**Authorship**

This study was conceived and designed by KIL and JSA. Preparation of and access to
specimens as well as review of the manuscript was provided by CFM. KIL provided additional
preparation, analysis, and interpretation of data. SLM connected KIL with Dragonfly ORS and
contributed to interpretation of the data. The manuscript was written by KIL with assistance from JSA.

**Conflict of Interest**

The authors are not aware of any conflict of interest.

**Additional Information**

Supplementary information is available for this paper. Correspondence and requests for
materials should be addressed to KIL.

[revised manuscript text omitted]

*10.4138/2076*
42. Tibert NE, Scott DB. 1999 Ostracodes and Agglutinated Foraminifera as Indicators of
Paleoenvironmental Change in an Early Carboniferous Brackish Bay, Atlantic Canada. *Palaios*
**14**, 246. (doi:10.2307/3515437)
43. Lebedev OA, Coates MI. 1995 The postcranial skeleton of the Devonian tetrapod *Tulerpeton*
*curtum* Lebedev. *Zool. J. Linn. Soc.* **114**, 307–348. (doi:10.1111/j.1096-3642.1995.tb00119.x)
44. Lombard RE, Bolt JR. 1995 New Primitive Tetrapod *Whatcheeria Deltae* From Lower
Carboniferous Iowa. *Palaeontology.* **38**.
45. Bishop PJ, Walmsley CW, Phillips MJ, Quayle MR, Boisvert CA, McHenry CR. 2015 Oldest
pathology in a tetrapod bone illuminates the origin of terrestrial vertebrates. *PLoS One* **10**,
e0125723. (doi:10.1371/journal.pone.0125723)
46. Smithson TR. 1985 The morphology and relationships of the Carboniferous amphibian
*Eoherpeton watsoni* Panchen. *Zool. J. Linn. Soc.* **85**, 317–410. (doi:10.1111/j.1096-
3642.1985.tb01517.x)
47. Lanyon LE, Goodship AE, Pye CJ, MacFie JH. 1982 Mechanically adaptive bone remodelling.
*J. Biomech.* **15**, 141–154. (doi:10.1016/0021-9290(82)90246-9)
48. Carter DR, Orr TE, Fyhrie DP. 1989 Relationships between loading history and femoral
cancellous bone architecture. *J. Biomech.* **22**, 231–44. (doi:10.1016/0021-9290(89)90091-2)
49. Canoville A, Laurin M. 2009 Microanatomical diversity of the humerus and lifestyle in
lissamphibians. *Acta Zool.* **90**, 110–122. (doi:10.1111/j.1463-6395.2008.00328.x)
50. Sanchez S, Germain D, De Ricqlès A, Abourachid A, Goussard F, Tafforeau P. 2010 Limb-bone
histology of temnospondyls: Implications for understanding the diversification of
palaeoecologies and patterns of locomotion of Permo-Triassic tetrapods. *J. Evol. Biol.* **23**, 2076–
2090. (doi:10.1111/j.1420-9101.2010.02081.x)
51. Quemeneur S, de Buffrénil V, Laurin M. 2013 Microanatomy of the amniote femur and
inference of lifestyle in limbed vertebrates. *Biol. J. Linn. Soc.* **109**, 644–655.
(doi:10.1111/bij.12066)

52. Dunmore CJ, Kivell TL, Bardo A, Skinner MM. 2019 Metacarpal trabecular bone varies with
distinct hand-positions used in hominid locomotion. *J. Anat.* **235**, 45–66.
(doi:10.1111/joa.12966)

[revised manuscript text omitted]

Appendix D

Response to reviewer 1

Comments to the Author(s)

Dear authors,

I really want to congratulate you for this study.

Although it is now well known that the first tetrapods were aquatic, we still see too many studies considering the appearance of limb bones correlated with a terrestrial lifestyle. Your work is a great step toward a better comprehension of the lifestyle in the first tetrapods.

I have just a few minor comments and I corrected some minor typos in the pdf file.

In text responses/additions are highlighted in bold.

The only concern about statistical analyses is about the principal component analysis. It is usually better to use phylogenetic PCA in order to correct phylogenetic effects but in this case, it will be difficult to use because of the uncertainty of the phylogenetic position of the fossil specimens. I just suggest you to explain why you did not use phylogenetic PCA.

We did not use a phylogenetic PCA because of poorly resolved early tetrapod phylogenies and high lever taxonomic ID's. Added a sentence into the statistics section addressing this.

I am especially grateful to the authors for their supplemental information concerning the step-by-step fossil bone segmentation method. This is really important to allow replication and still too rare in publications.

It would be good to cite the old literature about bone functional adaptation, first described by Roux (1885) and the 'Wolff law' (Wolff 1892).

Roux, Wilhem. (1885). Beitrage zur Morphologie der funktionellen Anpassung. Archives of Anatomy and Physiology, 9, 120–185.

Wolff, Julius. (1892). Das Gesetz der Transformation der Knochen. Berlin : A. Hirchwild.

Done, Thank-you!

p.8 l.135. You write that terrestrial animals have thicker layers of cortical bone but this is not totally true. In small terrestrial species, cortical bone tends to be thin whereas it tends to be thicker in heavy ones or in amphibious and aquatic taxa with a passive buoyancy control.

We corrected our language to compactness rather than thickness.

p.12 l.231. It may be phylogenetic too

True, we added that point. Again, this is hard to account for because of reasons previously outlined (lack of closely related animals, poorly resolves phylogenies)

Finally, the method you developed will be very useful to assess the lifestyle of extinct taxa, not only for the first tetrapods but also many other taxa. I am sure this article will be highly cited in the future.

It is a very interesting and important contribution about the paleobiology of the early tetrapods from Devonian of Nova Scotia. It is a very welcome voice which increase our understanding of the transition from water to land, the

problem which is still not fully clear. Moreover, authors are using an absolutely new method for paleontology, namely analysis of the orientation of the trabeculae. It is a very promising, non-invasive idea, complementary to early known method of the analysis of the 2D microanatomy, which, according, to the preliminary results really can help to work on the limited material without destroying it.

Although the issue is really interesting and deserves to be published in Royal Society Open Science, I have still found few problems which should be assessed before final acceptance of the paper.

- 1 The first is a structure of the paper. As I have checked others paleo-papers published in the journal, usually they have a standard structure with four main parts: introduction, material and methods, results and discussion. In your paper there is no clear material and method chapter – you have background which somehow provide information typical for M&M part, but in my opinion should be a little bit more structured.

Moved Materials and methods into the main body of the paper.

Your results part includes also few interpretations, which in standard structure should be in discussion.

Moved and reworked

- 2 The second important issue is terminology – you are using a term internal (trabecular) bone microanatomy – it is not a common definition used in paleohistology. For me it is not clear what do you mean.

Specified trabecular bone within the medullary cavity.

In mammal's histology, in long bones usually trabecular bone is an internal bone (sensu inside the bone). In early tetrapods trabecular bone it is a bone which fills the medullary region and usually is secondary, but also may build the periosteal domain (cortex, and here could be secondary or primary). Moreover, the border between medullary region and cortex is not sharp – mostly we have there wide perimedullary region. Doing an analysis in your study of the internal trabecular microstructure you mean only bone which fills medullary region or the entire cross section (2D – it is what bone profiler is doing) or bone (3d). If only inside, where is the border between periosteal domain and trabecular bone.

We used an automated Kholer function for isolating trabecular bone, the remaining bone was designated cortical, this is in the materials and methods section. It's true, I did not include periosteal.

You need to explain these details.

Moreover it will be easier for readers to follow your results and conclusions if in the introduction you will provide the basic information how the bone microstructure is connected with the mode of life (how can you deduce about mode of life based on the bone microstructure?).

We provided more information about bone microstructure and mode of life in the introduction, and how we interpret trabecular orientation and alignment according to Bone Functional Adaptation

For more information about paleohistological terminology you can check:

Francillon-Vieillot, H., V. de Buffrénil, J. Castanet, J. Géraudie, F.J. Meunier, J.Y. Sire, L. Zylberberg, and A. de Ricqlès. 1990. Microstructure and mineralization of vertebrate skeletal tissues; pp. 471–530 in J.G. Carter (ed.), *Skeletal Biomineralization: Patterns, Processes and Evolutionary Trends*, Vol. I. Van Nostrand Reinhold, New York.

- 3 The sectioning planes for microanatomy in 2d – it is important for partially preserved bones. For early tetrapods the midshaft (histological midshaft) is very short and even a small shifting of the sectioning plane in the proximal or distal direction may results in different microanatomical picture (and then different interpretation, mostly in favor of aquatic environment). I think you can discuss the influence of the sectioning plane on your results a little bit more (you did this for one bone very short, but others I am not sure if you are in midshaft).

We added details about how the orientation was determined, see materials and methods, “Two-dimensional analysis”

- 4 References – in some point your statements are too arbitral. You need more references to support given opinion. Moreover, I think that useful references for you will be:

We added more references where indicated in text. In one place we specified we were writing about our own observations.

Houssaye, A., Sander, M. P., and Klein, N. (2016). Adaptive patterns in aquatic amniote bone microanatomy - More complex than previously thought. *Integrative and Comparative Biology* 2016:1–21. DOI:10.1093/icb/icw120.

This was very useful! Thank you!

Sanchez, S., Tafforeau, P., Ahlberg, P. (2014). The humerus of Eusthenopteron: a puzzling organization presaging the establishment of tetrapod limb bone marrow. *Proceedings of the Royal Society of London. Biological Sciences*, ISSN 0962-8452, E-ISSN 1471-2954, Vol. 281, no 1782, p. 20140299

Sanchez, S., Dupret, V., Tafforeau, P., Trinajstić, K., Ryll, B. et al. (2013). 3D Microstructural Architecture of Muscle Attachments in Extant and Fossil Vertebrates Revealed by Synchrotron Microtomography. *SPLoS ONE*, vol. 8, ss. e56992- DOI

Sanchez, S., Tafforeau, P., Clack, J., Ahlberg, P. (2016). Life history of the stem tetrapod *Acanthostega* revealed by synchrotron microtomography. *Nature*, vol. 537, ss. 408+ DOI

Qu, Q., Sanchez, S., Zhu, M., Blom, H., Ahlberg, P. (2017). The origin of novel features by changes in developmental mechanisms: ontogeny and three-dimensional microanatomy of polyodontode scales of two early osteichthyans. *Biological Reviews*, vol. 92, ss. 1189-1212.

Especially the first is important, as authors in that paper show that conclusions about mode of life only based on the 2d microanatomy are not always very easy. Thus, your method of the analysis of the “trabecular” orientation is very important as a supporting source of information. Other papers are not dealing directly with the same topic as yours, but I think in introduction it will be nice to have short overview what was done up to date, as few attempts to use 3d modelling for the life reconstruction of early “tetrapods” are also known.

Added more details about two of the studies involving modeling joint mobility and humeral shape.

5. It will be nice to have a scale bars in your figures – I think that you have a scale bars, but there is no information what does it mean...

Added what the scale bars represents into the captions.

Finally, I would like to recommend the paper for publication after moderate revision. Please see attached pdf with my comments. Should the authors have any questions regarding my comments, they are welcome to contact me.

Appendix E

Response to reviewer 2

Reviewer: 2

Comments to the Author(s)

Lennie et al. investigates trabecular and cortical thickness as potential indicators of terrestriality. The authors use a modern sample with known habitat preferences and compare these modern analyses to fossils from Blue Beach, Nova Scotia. These fossils represent some of the earliest tetrapods and as such represent important data points in our understanding of vertebrate terrestrialization. The authors do establish the importance of finding more direct evidence of terrestriality than previous methods and include a very important dataset of fossils. I do believe these are important data and should be published, however I think there are substantial changes to the manuscript, interpretations, and perhaps even the sampling that are required prior to accepting this manuscript.

I have made some more specific comments in my marked version of the manuscript and I include some general themes from my comments here.

In-text responses/additions/changes are in bold

In the introduction to microstructure as a proxy for habitat (i.e. aquatic vs. terrestrial), there are two important points missing. First, I think it is important that the authors discuss the variability in aquatic indicators from modern taxa. There really is no ‘smoking gun’ in bone microstructure and the modern approach is to tackle this question looking a suite of features. This study does just that which I think is important, but I would urge more caution when describing what boney signals indicate. See discussion in Plasse et al. 2019 and Houssaye et al. 2016.

The importance of looking at multiple features and that some aquatic indicators may be associated with secondary return to water and offsetting buoyancy is discussed

Second, I would recommend including more sources describing experimental data on the effects of loading on trabeculae and cortical thickness. This would bolster the support for these features as possible indicators for terrestriality.

We have added references to the following additional studies:

Pontzer H, Lieberman DE, Momin E, Devlin MJ, Polk JD, Hallgrímsson B, et al. Trabecular bone in the bird knee responds with high sensitivity to changes in load orientation. *J Exp Biol.* 2006 Jan 1;209(1):57–65.

Barak MM, Lieberman DE, Hublin JJ. A Wolff in sheep’s clothing: Trabecular bone adaptation in response to changes in joint loading orientation. *Bone.* 2011;49(6):1141–51.

Text highlighting these studies was added to the introduction

The authors refer to their methodology as novel, however, I am not entirely clear how it is new. Is it the more holistic approach to trabecular anisotropy? The novel aspect of this method needs to be more clearly stated and if it is indeed novel, it requires in-depth descriptions of the method.

Yes, the novel part is the whole-bone approach. Clarified in the significance statement

There are many issues with the extant sample used in this study. It is unclear why all of the extant specimen with CT data were not run through all of the analyses.

Figure 1 has been updated to include more of the extant specimens that were used. The supplemental data contains all of the specimens, extant and extinct, that were used in the

analysis. An explanation of problems surrounding *Cryptobranchus*, *Elseya*, and *Caretta* was added to the methods section.

For instance, why are only the platypus and cat used in cortical thickness comparisons?

Data from the other comparative animals are included in supplemental material. Figure 1 was updated to include more.

There is no consistency between the extant sample included in the writing, the figures, and the table
Rectified.

Furthermore, the designation of aquatic vs. semi-aquatic is not consistent and the inclusion of semi-aquatic.

We changed “aquatic” animals to semi-aquatic (excluding *Cryptobranchus* and *Eusthenopteron*)

I would recommend increasing the sample especially since there are semi-aquatic taxa included.

We agree, and we hope this will be the basis for future research. We had acquired additional comparative specimens and planned for additional analyses to possibly be added after review but the pandemic intervened and this work will not be possible for the foreseeable future.

The inclusion of *Eusthenopteron* is interesting, however, it is never discussed in the text.

Added to results and discussion

The conclusion would be more robust if there were statistical analyses, even simple ones, to confirm their results.

We agree, but our sample size was very small, and with the data we have any statistical result would not be very powerful, increasing the sample size and dataset will (hopefully) be the subject of subsequent projects.

To me, there are some interpretations that I don't entirely agree with, but it is hard to say for sure since as presented, the interpretations are a matter of judgement.

Again we agree, especially for the anisotropy histograms, exporting the data effectively from Dragonfly will help to make more meaningful quantitative assessments in future studies.

Finally, I think caution in grouping all of these fossils together should be a bit more prevalent in the conclusions. The authors do acknowledge the fact that there were likely many contemporaneous levels of terrestriality, and I think they're results are actually strong on that point. To me, it appears that there are some subtle differences, but as stated above, it is difficult to say without statistics. It is more likely than not that these animals were doing a variety of actions with their limbs and so I would caution making one designation for all early tetrapods based on these data.